# PIXEL-PERFECT PUPPETRY: PRECISION-GUIDED ENHANCEMENT FOR FACE IMAGE AND VIDEO EDITING

**Yan Li**[1]  **Zhenyi Wang**[2]  **Guanghao Li**[3]  **Wei Xue**[1]  **Wenhan Luo**[1*]  **Yike Guo**[1*]

[1]The Hong Kong University of Science and Technology, [2]University of Central Florida,
[3]Tsinghua Shenzhen International Graduate School, Tsinghua University

## ABSTRACT

Preserving identity while precisely manipulating attributes is a central challenge in face editing for both images and videos. Existing methods often introduce visual artifacts or fail to maintain temporal consistency. We present **FlowGuide**, a unified framework that achieves fine-grained control over face editing in diffusion models. Our approach is founded on the local linearity of the UNet bottleneck's latent space, which allows us to treat semantic attributes as corresponding to specific linear subspaces, providing a mathematically sound basis for disentanglement. FlowGuide first identifies a set of orthogonal basis vectors that span these semantic subspaces for both the original content and the target edit, a representation that efficiently captures the most salient features of each. We then introduce a novel guidance mechanism that quantifies the geometric alignment between these bases to dynamically steer the denoising trajectory at each step. This approach offers superior control by ensuring edits are confined to the desired attribute's semantic axis while preserving orthogonal components related to identity. Extensive experiments demonstrate that FlowGuide achieves state-of-the-art performance, producing high-quality edits with superior identity preservation and temporal coherence. Our code is available at: `https://github.com/yl4467/flow_edit`.

## 1 INTRODUCTION

Face attribute editing has emerged as an essential task in computer vision, with applications ranging from film production to virtual reality, social media content, and digital avatars (Zhan et al., 2023; Kim et al., 2023; Yao et al., 2021; Zhang et al., 2018a; Zhu et al., 2020; Ye et al., 2025a). This task encompasses both face image editing (FIE) and face video editing (FVE), each presenting unique challenges. FIE demands precise attribute manipulation while preserving identity and avoiding unintended artifacts (Shen et al., 2020; Wang et al., 2022). FVE inherits these challenges but adds the critical requirement of temporal consistency across frames (Wang et al., 2024; Ceylan et al., 2023). Current methods often struggle to satisfy all these constraints simultaneously. To address this, we propose a unified, pixel-level solution for both FIE and FVE that enhances editing precision while maintaining identity and temporal coherence.

Early approaches to face editing predominantly relied on GAN-based methods (Tzaban et al., 2022; Patashnik et al., 2021; Karras et al., 2019; Shen et al., 2020), which utilize pre-trained StyleGAN models and GAN inversion techniques (Karras et al., 2020; Xia et al., 2022). These methods map input images or video frames into a latent space where edits can be applied. However, the quality of edits heavily depends on the accuracy of GAN inversion, which often struggles to faithfully reconstruct the original input, leading to identity loss and editing artifacts (Preechakul et al., 2022). For video editing, GAN-based methods face additional challenges in maintaining temporal coherence, often resulting in flickering or inconsistent edits across frames.

Recent advances in diffusion models have shown superior performance in face editing tasks (Batifol et al., 2025; Kim et al., 2023; Preechakul et al., 2022; Zhang et al., 2023; Croitoru et al., 2023). These methods perform editing as a conditional generation process, where target attributes are progressively introduced during the denoising steps. While diffusion models offer better reconstruction

---

*Corresponding author

quality and more stable generation compared to GANs, they still lack precise control over the editing process (Zhao et al., 2024; Yu et al., 2023). Without proper constraints, introducing target attributes can inadvertently affect other facial features, identity, or background elements—a problem that becomes particularly pronounced in video editing where such errors accumulate across frames.

To address these limitations, we propose **FlowGuide**, a unified framework that achieves precise face editing by introducing a novel guidance mechanism operating within the diffusion model's latent space. Our approach is founded on the local linearity of the UNet bottleneck's latent space (Park et al., 2023; Kwon et al., 2022), which allows us to treat semantic attributes as corresponding to specific linear subspaces. To disentangle identity from attributes, our *Latent Basis Extraction (LBE)* module first identifies a set of orthogonal basis vectors that span these key semantic directions for both original and edited content. The core of our method is an *Implicit Basis Guidance (IBG)* mechanism that quantifies the semantic change by measuring the geometric alignment between these two sets of basis vectors. This alignment score informs a corrective update to the predicted noise at each denoising step, effectively steering the generation trajectory along the desired attribute's semantic axis while preserving components orthogonal to it, which correspond to identity and other preserved features. This ensures precise, localized edits for images and naturally extends to temporally coherent modifications for videos.

We summarize the contributions of our proposed method as follows:

- We propose FlowGuide, a unified framework for face image and video editing that introduces a novel guidance mechanism to achieve precise attribute control in diffusion models.
- We treat semantic attributes as linear subspaces within the UNet bottleneck's latent space, designing a Latent Basis Extraction (LBE) module to identify orthogonal basis vectors that span these subspaces to isolate the identity from the attributes in the latent space.
- We introduce an Implicit Basis Guidance (IBG) mechanism that computes the geometric alignment between these bases to dynamically steer the denoising trajectory, which confines edits to the target attribute's semantic axis while preserving the identity.
- Extensive experiments demonstrate that FlowGuide achieves state-of-the-art editing quality, with superior identity preservation, attribute modification, and temporal coherence.

## 2 RELATED WORK

### 2.1 INVERSION-BASED IMAGE EDITING

Inversion-based editing in diffusion models began with deterministic methods like DDIM inversion (Song et al., 2020). To improve identity preservation, subsequent optimization-based approaches like Null-text Inversion (NTI) (Mokady et al., 2023) and Prompt Tuning Inversion (PTI) (Roich et al., 2022) fine-tuned text embeddings, though at a significant computational cost. To address this inefficiency, a variety of optimization-free methods were developed. Negative Prompt Inversion (NPI) (Miyake et al., 2023) and ProxNPI (Han et al., 2024) bypass direct optimization of embeddings, while others like PnP Inversion (Ju et al., 2023) and Noise Map Guidance (NMG) (Cho et al., 2024) use guidance or directly incorporate reconstruction differences into the editing update.

More recent works have explored alternative strategies beyond direct deterministic inversion. For instance, Edit Friendly (EF) (Huberman et al., 2024) and its successor LEDITS++ (Brack et al., 2024) employ random inversion to achieve good reconstruction without requiring attention map adjustments. Concurrently, methods like h-Edit (Nguyen et al., 2025) have introduced hierarchical frameworks for more granular semantic control. Despite this progress, most training-free methods remain limited to text-guided editing and struggle to achieve precise control over attributes without affecting non-target regions. Our work addresses this by introducing a guidance mechanism that operates on the fundamental geometric structure of the latent space, enabling more precise and disentangled control.

### 2.2 FACE VIDEO EDITING

Face video editing (FVE) aims to modify facial attributes in videos while preserving identity and temporal consistency. Early FVE methods often relied on GANs, particularly StyleGAN (Karras

et al., 2019). Approaches like StyleCLIP (Patashnik et al., 2021) and Stitch it in Time (STIT) (Tzaban et al., 2022) perform GAN inversion to project video frames into StyleGAN's latent space for editing. However, these methods are often limited by the quality of GAN inversion Patashnik et al. (2021); Shen et al. (2020), which can lead to identity loss and artifacts, and they struggle to maintain temporal coherence across frames Preechakul et al. (2022).

Recent works have further advanced diffusion-based video editing through various approaches Geyer et al. (2023); Yang et al. (2023); Qi et al. (2023); Ouyang et al. (2024); Kara et al. (2024); Anand et al. (2025); Li et al. (2025); Ye et al. (2025b); Kong et al. (2024). RAVE (Kara et al., 2024) introduces randomized noise shuffling for fast and consistent editing but focuses primarily on semantic scene editing rather than fine-grained facial control. IP-FaceDiff (Anand et al., 2025) specifically targets identity preservation in facial videos, while Qffusion (Li et al., 2025) employs quadrant-grid attention learning for controllable portrait editing. V-LASIK (Shalev-Arkushin et al., 2024) addresses the specific challenge of consistent glasses removal using synthetic data. Other notable advances include (Liao & Deng, 2023) extends ControlNet to video generation with cross-frame attention, and (Lu et al., 2024) performs high-fidelity video editing via multi-source diffusion. While effective, diffusion-based approaches often lack fine-grained control, leading to unintended modifications of non-target attributes. Our work addresses this limitation by introducing a mechanism for precise, guided control within the diffusion framework, ensuring that edits are confined to the desired attributes while preserving identity and temporal stability.

## 3 METHODOLOGY

### 3.1 PRELIMINARIES: DIFFUSION-BASED EDITING

Let $X_0$ represent an input frame. Our method supports processing multiple frames simultaneously; for simplicity, we use $X_0$ to denote the input in the following sections. Our dual-path framework, and diffusion-based editing in general, operates by first inverting $X_0$ into a noisy latent representation, which is then denoised. Each frame is processed independently through this pipeline. We denote variables associated with the identity-preserving **reconstruction path** with a superscript $r$ and variables for the **editing path** with a superscript $c$.

The **inversion process** is a deterministic DDIM-based procedure that progressively adds noise to create the starting latent for the reconstruction path, $X_T^r$. The transition from a less noisy latent $X_{t-1}^r$ to a more noisy latent $X_t^r$ under the original condition $c_{edit}$ is modeled as:

$$q(X_t^r|X_{t-1}^r, c_{edit}) = \mathcal{N}(X_t^r; \mu_\theta(X_{t-1}^r, t, c_{edit}), \sigma_t^2 \mathbf{I}) \tag{1}$$

where the mean is a function of the predicted noise $\epsilon^r(X_{t-1}^r, t, c_{edit})$: $\mu_\theta(X_{t-1}^r, t, c_{edit}) = 1/\sqrt{\alpha_t}(X_{t-1}^r - 1 - \alpha_t/\sqrt{1 - \bar{\alpha}_t}\epsilon^r(X_{t-1}^r, t, c_{edit}))$, $\alpha_t$ is the noise schedule coefficient.

The **denoising process** generates the edited image by iteratively removing noise, guided by a target condition $\mathcal{C}^c$. The editing path starts from the same noisy latent as the reconstruction path, $X_T^c = X_T^r$. The denoising step for the editing path is defined as:

$$p_\theta(X_{t-1}^c|X_t^c, \mathcal{C}^c) = \mathcal{N}(X_{t-1}^c; \mu_\theta(X_t^c, t, \mathcal{C}^c), \Sigma_\theta(X_t^c, t, \mathcal{C}^c)) \tag{2}$$

where the mean $\mu_\theta$ is a function of the noise $\epsilon^c(X_t^c, t, \mathcal{C}^c)$ predicted under the target condition.

To improve consistency, the **edit-friendly guidance** (Huberman et al., 2024) can be introduced into the denoising process, which explicitly links the reconstruction and editing paths. The intuition is to ground the editing process in the reconstruction process to prevent it from deviating too far. The edit-friendly guidance is defined as:

$$X_{t-1}^c = X_{t-1}^r - \mu_\theta(X_t^r, t, c_{edit}) + \mu_\theta(X_t^c, t, \mathcal{C}^c) \tag{3}$$

While this technique enforces a strong structural prior from the reconstruction path, it often proves to be too restrictive. The guidance is not adaptive; it does not dynamically measure how much the edit should differ from the original. Consequently, such methods often lack the fine-grained control needed to robustly preserve identity while making significant, targeted attribute changes. This fundamental limitation motivates our work, which introduces a more advanced, adaptive guidance mechanism to enhance editing accuracy.

## 3.2 OVERVIEW OF FLOWGUIDE

Our method operates through two parallel processes: reconstruction and editing to achieve precise attribute manipulation while preserving identity, as illustrated in Figure 1. Both processes invert the input frames to noisy latents but use different conditions during the denoising phase. The reconstruction process uses the original conditions $(c_{org}, \mathcal{C}^r)$ to establish an identity-preserving reference path. Concurrently, the editing process uses the target conditions $(c_{edit}, \mathcal{C}^c)$ to introduce the desired attribute modifications.

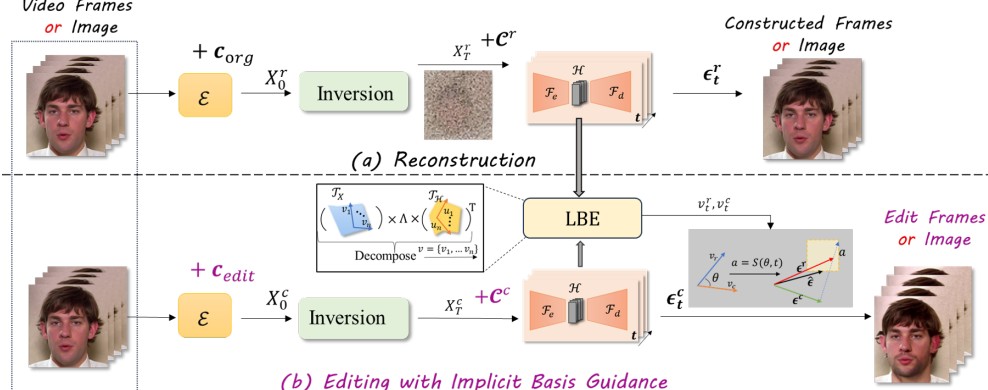

Figure 1: **The framework of proposed FlowGuide.** (a) The reconstruction process shows how original frames are inverted to noisy latents $X_T^r$ with original condition $c_{org}$, then denoised back with condition $\mathcal{C}^r$, establishing the baseline for identity preservation. (b) The editing process of our method: first invert latent representations to $X_T^c$ with editing condition $c_{edit}$, during denoising with target condition $\mathcal{C}^c$, we extract latent basis vectors from the UNet bottleneck layer, and apply implicit basis guidance to ensure edits are confined to target attributes.

During each step of the parallel denoising, our Latent Basis Extraction (LBE) module (Section 3.3) is applied to the UNet bottleneck of both paths. This yields two sets of basis vectors: $\mathcal{V}^r$ for the original content and $\mathcal{V}^c$ for the edited content. Our key contribution, the Implicit Basis Guidance (IBG) mechanism (Section 3.4), then computes the similarity between $\mathcal{V}^r$ and $\mathcal{V}^c$ to quantify semantic change. This similarity dynamically steers the denoising direction of the editing path, ensuring modifications are confined to target attributes while preserving all other characteristics from the reconstruction baseline.

This dual-process framework naturally extends to both single images and video sequences, where temporal consistency emerges from the coherent application of basis guidance across frames. Detailed inversion procedures for image and video modalities are provided in Appendix E.2 and F.1, respectively.

## 3.3 LATENT BASIS EXTRACTION

Building on the dual-process framework described above, the noisy representations $X_T^r$ and $X_T^c$ are fed into a pre-trained UNet $\mathcal{F}$ to predict the noise of each frame. Within this architecture, we use $\mathcal{F}_e$ and $\mathcal{F}_d$ to denote the encoder and decoder components of the UNet, respectively. Since the *process* of extracting the latent basis is the same for both paths (though the resulting bases $V^r$ and $V^c$ differ), we use $X_T^c$ as an example for simplicity. To streamline the presentation, we let $\mathcal{X}$ represent $X_t^c$, $\mathcal{H}$ denote the latent variable, and $\mathcal{C}$ represent $\mathcal{C}^c$ at time step $t$.

The latent variable $\mathcal{H}$ in the bottleneck layer of the U-Net has been shown to exhibit a locally linear structure (Kwon et al., 2022), which makes it suitable for using the Euclidean metric to measure changes in $\mathcal{H}$ (Kim et al., 2023). In the denoising process, the transformation from the input representations to the latent space can be expressed as $\mathcal{F}_e : \mathcal{X}, \mathcal{C} \to \mathcal{H}$, where $\mathcal{F}_e$ maps the input $\mathcal{X}$ and the editing conditions $\mathcal{C}$ to the latent variable $\mathcal{H}$. However, since $\mathcal{X}$ contains a lot of information unrelated to the specific editing direction, the variability it introduces into $\mathcal{H}$ might not align with the desired editing directions. To overcome this issue, we focus primarily on how $\mathcal{C}$ (the editing

condition) influences $\mathcal{H}$, effectively isolating the impact of the target attribute from other unrelated aspects of $\mathcal{X}$. This approach enables us to better control the editing process by only adjusting the components of $\mathcal{H}$ that are relevant to the intended changes, ensuring more precise and consistent video edits.

Since the video editing process incorporates the additional condition $\mathcal{C}$ into the denoising steps, $\mathcal{C}$ directly influences key features in the latent space $\mathcal{T}_{\mathcal{H}}$, where $\mathcal{T}_{(.)}$ denotes the vector space. Therefore, our goal is to identify the local latent vectors $\mathcal{V} = \{v_1, \ldots, v_n\} \in \mathcal{T}_{\mathcal{C}}$ that exhibit significant variability within the tangent space of the latent variable $\mathcal{H}$, denoted as $\mathcal{T}_{\mathcal{H}}$. By focusing on these local latent vectors, we can effectively capture the key aspects of the editing direction that drive changes in the latent space, ensuring that the manipulation of the video aligns with the intended attribute modifications while preserving other important details such as identity and background. We provide a detailed analysis of the impact of the latent basis on the editing process in Appendix C.

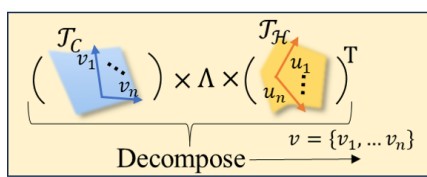

Figure 2: The illustration of extracting the latent basis.

The linear relationship between $\mathcal{C}$ and $\mathcal{H}$ can be expressed as a linear map: $\mathcal{T}_{\mathcal{C}} \rightarrow \mathcal{T}_{\mathcal{H}}$. This linear transformation is described by the Jacobian matrix $J_{\mathcal{C}}$, which captures how a vector $v \in \mathcal{T}_{\mathcal{C}}$ is mapped to a vector $u \in \mathcal{T}_{\mathcal{H}}$ through the relation $u = J_{\mathcal{C}} v$. Given the local linearity of $\mathcal{H}$ in the latent space, the pullback of $\mathcal{H}$ allows us to assign a meaningful geometric structure to $\mathcal{C}$, enabling more precise control over the editing process by understanding how changes in $\mathcal{C}$ affect the latent space $\mathcal{H}$, the norm of $v$ can be measured:

$$||v||_{pb}^2 = <u, u>_{\mathcal{H}} = v^\top J_{\mathcal{C}}^\top J_{\mathcal{C}} v \tag{4}$$

where $<u, u>_{\mathcal{H}} = u^\top u$ is the dot product of $u$ defined in the Euclidean space with the local linearity of $\mathcal{H}$.

The vectors $\mathcal{V} = \{v_1, \ldots, v_n\} \in \mathcal{T}_{\mathcal{C}}$ that maximize $||v||_{pb}^2$ can be derived through the singular value decomposition (SVD) of the Jacobian matrix $J_{\mathcal{C}} = U \Lambda \mathcal{V}^\top$, as illustrated in Figure 2. Here, $\mathcal{V} = \{v_1, \ldots, v_n\}$ represents the right singular vectors of $J_{\mathcal{C}}$, $U = \{u_1, \ldots, u_n\} \in \mathcal{T}_{\mathcal{H}}$ represents the left singular vectors, and $\Lambda$ is a diagonal matrix of singular values, it has $J_{\mathcal{C}} v_i = \Lambda_i u_i$. The extracted latent basis vectors $\mathcal{V} = \{v_1, \ldots, v_n\}$ correspond to directions in the latent space that are highly responsive to the conditions encoded in $\mathcal{C}$, offering key insights into how the video editing process responds to specific attributes. Henceforth, we obtain the latent basis responses corresponding to the conditions $\mathcal{C}^r$ and $\mathcal{C}^c$, denoted as $\mathcal{V}^r = \{v_1^r, \ldots, v_n^r\}$ and $\mathcal{V}^c = \{v_1^c, \ldots, v_n^c\}$, respectively. Having extracted these basis vectors, the next critical step is to use them to guide the denoising process by our Implicit Basis Guidance, as detailed in the following section.

## 3.4 Implicit Basis Guidance

To quantify the degree of alignment between the original and manipulated conditions, we measure the similarity between the latent basis vectors $\mathcal{V}^r$ and $\mathcal{V}^c$. This similarity provides a means to assess the extent of changes introduced during the editing process. Among various similarity metrics (Pearson correlation, Spearman correlation, etc.), we adopt cosine similarity to measure the relationship between $\mathcal{V}^r$ and $\mathcal{V}^c$:

$$\Phi_{\mathcal{C}}(\mathcal{V}^r, \mathcal{V}^c) = \cos^{-1}(\phi)/\pi, \ \cos(\phi) = \frac{1}{n} \sum_{i=1}^{n} \frac{v_i^r v_i^c}{||v_i^r|| ||v_i^c||} \tag{5}$$

The choice of cosine similarity is motivated by its ability to capture directional relationships between latent basis vectors while remaining robust to magnitude variations, which is particularly important for measuring semantic changes in the latent space. We empirically validate this choice by comparing cosine similarity with Pearson correlation and Spearman correlation in Table 1, with detailed comparative analysis provided in Section 4.1.2.

The latent basis associated with different conditions is extracted as described in Section 3.3, and the similarity between $\mathcal{V}^r$ and $\mathcal{V}^c$ can be utilized to provide more precise guidance for video manipulation. We denote the computed similarity as $\xi = \Phi_{\mathcal{C}}(\mathcal{V}^r, \mathcal{V}^c)$, refer to Equation 5. This similarity $\xi$

serves as a key factor in adjusting the manipulation process, ensuring that only the target attributes are modified while preserving other important characteristics like identity and background. Given that the similarity $\xi = \Phi_\mathcal{C}(\mathcal{V}^r, \mathcal{V}^c)$ measures the impact of the conditions on the model, we propose using this similarity as guidance to regulate the denoising process.

To ensure that the magnitude of deviation between $\epsilon^c$ and $\epsilon^r$ is proportional to the similarity $a$, we use the similarity value to determine which regions should be edited. Specifically, when the latent bases are very similar (high $a$), only small regions should differ between the two paths; when the bases are dissimilar (low $a$), larger regions can be modified. To achieve this, we employ a dynamic threshold rather than a fixed one. We select the $1 - a$ quantiles from the matrix $|\epsilon^c - \epsilon^r|$ and denote the cutoff value as $\lambda$. This allows us to construct a binary mask and compute the final guided noise:

$$\mathcal{M} = |\epsilon^c - \epsilon^r| < \lambda, \; \hat{\epsilon} = \epsilon^c + \mathcal{M} \odot (\epsilon^r - \epsilon^c) \tag{6}$$

where $\hat{\epsilon}$ is the final noise prediction used in the denoising step, blending the editing noise $\epsilon^c$ with the reconstruction noise $\epsilon^r$ according to the mask $\mathcal{M}$. This method enables us to focus edits on regions with significant latent basis differences, effectively filtering out less relevant information to ensure the target attributes are modified while maintaining the integrity of non-target features.

## 4 EXPERIMENT

### 4.1 FACE IMAGE EDITING

#### 4.1.1 EXPERIMENT SETTING

**Dataset.** To evaluate the performance of face image editing, we select 500 images from the CelebA dataset Liu et al. (2015). We employ GPT-4o to generate comprehensive editing prompts encompassing five distinct editing tasks: "Add Sunglasses", "Add Makeup", "Age Progression", "Hair Color Modification", and "Add Smile". The detailed construction methodology for editing prompts is provided in the Appendix E.3. Furthermore, to assess the generalizability of our approach beyond facial editing, we conduct additional evaluations on the PIE-Bench dataset Ju et al. (2023) to measure general-purpose editing capabilities, the results on PIE Benchmark can refer to Appendix E.5.

**Baseline.** We compare our proposed method against state-of-the-art image editing approaches, including h-Edit Nguyen et al. (2025), NP Miyake et al. (2025), NMG Cho et al. (2024), EF Huberman et al. (2024), and PnP Inv Ju et al. (2023). To ensure fair and consistent evaluation across all methods, we employ p2p control to enhance reconstruction performance for each baseline.

**Metric.** For evaluation, we follow the evaluation setting in Nguyen et al. (2025), three main aspects are considered: 1) edited image quality, 2) editing effectiveness, 3) consistency between the original image and the edited image. To evaluate the edited image quality, we compute PSNR, LPIPS, and SSIM on non-edited regions. To measure the editing effectiveness, both standard CLIP similarity between the edited image and text and directional CLIP similarity between the edited image and text are used. To evaluate the consistency between the original image and the edited image, we compute DINO feature distance and the MSE distance between the original image and the edited image.

#### 4.1.2 QUANTITATIVE RESULTS

Quantitative results are presented in Table 1, comparing our method against five state-of-the-art baselines. We evaluate three variants of our model: one using cosine similarity (our primary proposal), and two others using Spearman and Pearson correlation for guidance. Both the cosine and Spearman variants demonstrate a superior trade-off between editing effectiveness (CLIP similarity) and identity preservation, significantly outperforming the Pearson variant, which produces overly aggressive edits that degrade identity. This outcome confirms our theoretical analysis (Section 3): angular and rank-based similarity metrics (Cosine, Spearman) better capture the geometric relationships in the latent space, providing more precise guidance than Pearson correlation, which is limited to linear relationships.

Across all methods, an inherent trade-off exists between editing strength and consistency. As illustrated in Figure 3, our cosine and Spearman-based FlowGuide variants achieve a more favorable balance than strong baselines like h-Edit, attaining higher quality and identity scores while remaining competitive on edit alignment. While the Spearman variant achieves the highest scores in quality

and consistency, the cosine variant provides a slightly better balance with edit strength, making it our recommended approach. Both demonstrate that our geometrically-grounded guidance mechanism enables more controlled and robust editing.

Table 1: The text-guided face image editing performance of different editing methods.

| Method | Edited Image Quality | | | Edited Performance | | Consistency | |
|---|---|---|---|---|---|---|---|
| | PSNR (↑) | LPIPS (↓) | SSIM (↑) | CLIP Sim (↑) | Local CLIP (↑) | DINO Dist (↓) | MSE Dist (↓) |
| EF Huberman et al. (2024) | 20.012 | 0.2028 | 0.7184 | 20.714 | 0.1225 | 0.0349 | 0.0109 |
| PnP Inv Ju et al. (2023) | 20.370 | 0.1343 | 0.7967 | 20.530 | 0.1296 | 0.0271 | 0.0106 |
| NMG Cho et al. (2024) | 14.679 | 0.3437 | 0.5673 | 21.666 | 0.1348 | 0.0831 | 0.0360 |
| NP Miyake et al. (2025) | 11.929 | 0.4747 | 0.4031 | 20.918 | 0.1409 | 0.1257 | 0.0665 |
| h-Edit Nguyen et al. (2025) | 22.078 | 0.1034 | 0.8341 | 19.707 | **0.1546** | 0.0193 | 0.0078 |
| FlowGuide (Pearson) | 16.988 | 0.2223 | 0.6988 | **22.157** | 0.1451 | 0.0539 | 0.0224 |
| FlowGuide (Spearman) | **24.129** | **0.0882** | **0.8642** | 17.831 | 0.1437 | **0.0161** | **0.0055** |
| FlowGuide (Cosine) | 23.160 | 0.0965 | 0.8448 | 19.391 | 0.1479 | 0.0166 | 0.0060 |

### 4.1.3 QUALITATIVE RESULTS

We visualize the face image editing results comparing our proposed method with baseline methods in Figure 3. Our method achieves superior editing quality and maintains better consistency between the original and edited images, though with slightly lower CLIP similarity between the edited image and text prompt. These visualization results align with the quantitative findings in Table 1, confirming that our method achieves more precise and consistent editing, thereby demonstrating the superiority of our proposed approach in face image editing. We provide additional visualizations of our method's face image editing capabilities in Appendix E.4 (Figure 8).

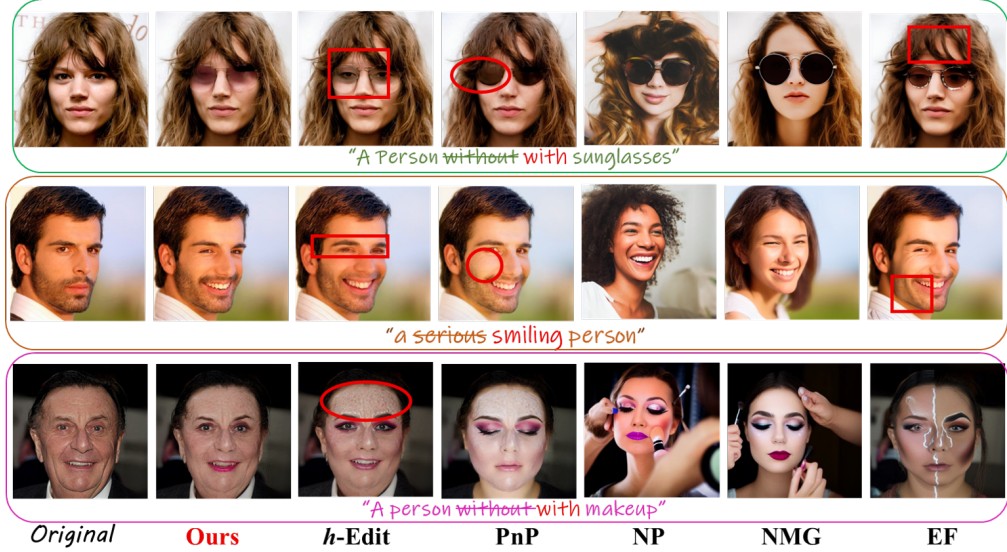

Figure 3: The comparison of the edited face image between our method and the baseline methods.

### 4.2 FACE VIDEO EDITING

### 4.2.1 EXPERIMENT SETTING

**Dataset.** We evaluate the performance of our proposed FlowGuide on real-world videos sampled from the HDTF dataset (Zhang et al., 2021) and the VoxCeleb dataset (Nagrani et al., 2017). Specifically, we randomly select 20 videos from each dataset, ensuring diversity across gender, age, and skin tones. Each video consists of hundreds of frames, from which we randomly sample 32 consecutive frames for each evaluation. The selected frames are aligned and cropped following the approach in (Tzaban et al., 2022; Kim et al., 2023), and subsequently resized to a resolution of $256 \times 256$.

**Baseline.** We compare our method extensively with several previous state-of-the-art baselines. We choose diffusion-based editing method DVA (Kim et al., 2023) and transformer-based method Latent-trans (Yao et al., 2021). For GAN-based methods, we choose STIT (Tzaban et al., 2022),

TCSVE Xu et al. (2022), PTI (Roich et al., 2022) and StyleCLIP (Patashnik et al., 2021). Some of the baseline methods are designed for image editing, we adapt them into the video editing paradigm (the details can refer to Appendix F.2). It is important to note that, for a fair comparison of the reconstruction abilities of different editing methods, the original videos are used solely as input.

**Metric.** For comprehensive evaluation of our proposed FlowGuide and the baseline methods, we utilize a range of evaluation metrics. For the evaluation of reconstruction performance, we use SSIM (Wang et al., 2004), LPIPS (Zhang et al., 2018b), MSE and FID. For time consistency evaluation of manipulated videos, we apply TL-ID and TG-ID (Tzaban et al., 2022). For evaluating video editing performance, we use the Identity Preservation Rate (IPR), Target Attribute Change Rate (TACR) (Yao et al., 2021), and CLIP score.

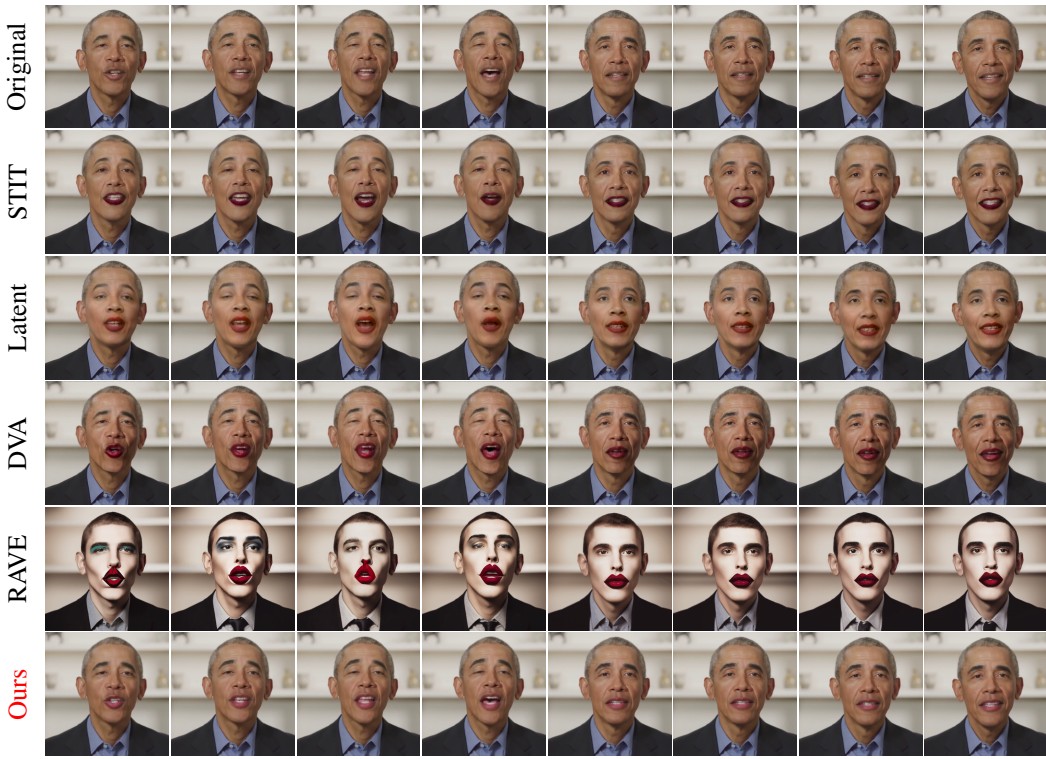

Figure 4: Comparison of editing performance of our FlowGuide to the previous video editing methods for editing direction 'Libstick'.

### 4.2.2 QUANTITATIVE RESULTS

To thoroughly evaluate the editing capabilities of our proposed FlowGuide compared to baseline methods, we choose two general editing directions ("Smiling", "Mustache"). We compute and report the average values of key evaluation metrics, such as Identity Preservation Rate (IPR), Target Attribute Change Rate (TACR), and CLIP score, for both our method and the baseline approaches. The results, summarized in Table 2, illustrate how effectively each method handles these editing tasks, offering insights into their relative performance across different editing scenarios. The reconstruction ability of different methods are presented in Appendix F.3.

As shown in Table 2, our proposed FlowGuide achieves the highest Identity Preservation Rate (IPR), highlighting its effectiveness in maintaining identity information during editing process. Notably, RAVE shows significantly lower performance compared to face-specific methods. This performance gap highlights a fundamental challenge: general video editing methods are designed for semantic scene editing and large-scale motion manipulation, where spatial consistency requirements are relatively relaxed. In contrast, face video editing demands extremely precise pixel-level consistency to the input video and fine-grained control over subtle facial attributes while maintaining identity. The human visual system is highly sensitive to facial inconsistencies, making it particularly challenging to apply general video editing approaches to face manipulation tasks. Additionally, our

Table 2: The editing ability of our FlowGuide and baselines on HDTF and VoxCeleb datasets. The reported values are the mean of two editing directions ("Smile", "Mustache").

| Method | HDTF | | | | | VoxCeleb | | | | |
|--------|------|------|------|------|------|------|------|------|------|------|
| | IPR (↑) | TACR (↓) | CLIP-Score (↑) | TL-ID (↑) | TG-ID (↑) | IPR (↑) | TACR (↓) | CLIP-Score (↑) | TL-ID (↑) | TG-ID (↑) |
| StyleCLIP | 0.8013 | 0.0329 | 0.7676 | 0.9997 | 0.9995 | 0.7051 | 0.0337 | **0.7670** | 0.9998 | 0.9993 |
| STIT | 0.8214 | 0.0341 | 0.7501 | 0.9866 | 0.9490 | 0.8131 | 0.0339 | 0.7383 | 0.9997 | 0.9994 |
| PTI | 0.7540 | 0.0327 | 0.7646 | 0.8238 | 0.8122 | 0.7140 | 0.0336 | 0.7627 | 0.7986 | 0.8047 |
| TCSVE | 0.9413 | 0.0342 | 0.7566 | 0.9864 | 0.9770 | 0.8723 | 0.0029 | 0.7218 | 0.9813 | 0.9077 |
| Latent-trans | 0.7515 | 0.0348 | 0.7450 | 0.9978 | 1.0000 | 0.7070 | 0.0335 | 0.7393 | 0.9999 | **1.0000** |
| DVA | 0.9244 | **0.0318** | 0.7685 | 1.0000 | 0.9977 | 0.8910 | 0.0341 | 0.7661 | 0.9999 | 0.9969 |
| RAVE | 0.7005 | 0.0338 | 0.7295 | 0.8621 | 0.7731 | 0.6812 | 0.0341 | 0.7301 | 0.8598 | 0.7684 |
| FlowGuide | **0.9667** | 0.0338 | **0.7777** | **1.0001** | **1.0000** | **0.9033** | 0.0335 | 0.7607 | **1.0000** | **1.0000** |

method demonstrates comparable temporal consistency to the baseline methods, further validating its robustness in preserving video quality.

### 4.2.3 QUALITATIVE RESULTS

Qualitative comparisons are presented in Figure 4, with additional results in Appendix F.9. The visualizations demonstrate our method's ability to perform precise, localized edits. As shown, FlowGuide effectively modifies the target attribute while preserving the subject's identity, non-target facial features, and the background. This high degree of control ensures the character's identity remains intact and the scene's original context is undisturbed.

To further showcase the robustness and generalizability of our method, we provide results for multiple, distinct edits on a single video in Appendices F.7. These examples highlight our model's ability to handle challenging, dynamic scenarios with intricate backgrounds, substantial head movements where many state-of-the-art methods falter. Our approach consistently retains the stylistic elements of the original

Table 3: The ablation results of our FlowGuide on HDTF dataset with two editing directions ("Smile" and "Mustache").

| Method | IPR (↑) | TACR (↓) | CLIP-Score (↑) | TL-ID (↑) | TG-ID (↑) |
|--------|---------|----------|----------------|-----------|-----------|
| w/o LBE | 0.9831 | 0.0331 | 0.7437 | 0.9925 | 0.9775 |
| w/o IBG | 0.9370 | 0.0337 | 0.7773 | 0.9770 | 0.8854 |
| w/o both | 0.8790 | 0.0309 | 0.7540 | 0.9590 | 0.8557 |
| FlowGuide | 0.9510 | 0.0329 | 0.7563 | 0.9986 | 0.9929 |

video, producing exceptionally natural edits that blend seamlessly with the original content. This ability to maintain coherence across diverse and challenging edits underscores the effectiveness of our guidance mechanism. We further report the computation efficiency of our method and the baseline methods in Appendix F.6.

### 4.3 ABLATION STUDY

We conduct an ablation study of the video editing tasks to analyze the contributions of our two core components: Latent Basis Extraction (LBE) and Implicit Basis Guidance (IBG), with results presented in Table 3 and Figure 5. First, we evaluate the importance of LBE by removing the module and computing similarity directly on the raw latent variables. This prevents the model from isolating attribute-specific features; the resulting guidance is too diffuse to apply the desired edit (low Target Attribute Change Rate), demonstrating that LBE is crucial for identifying the correct semantic directions for modification. A detailed analysis of the impact of the latent basis on the editing process is provided in Appendix C.

Next, we remove the IBG module while retaining LBE to assess its distinct role. Without IBG, the model correctly identifies what to change but lacks spatial control, applying edits indiscriminately across the entire frame. This

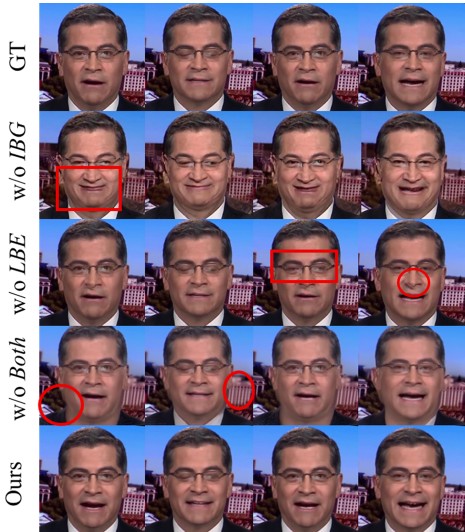

Figure 5: The ablation results of FlowGuide when apply editing direction:"smile".

leads to significant identity degradation (low IPR) and uncontrolled attribute changes, highlighting IBG's critical role in providing the spatial guidance necessary for localized edits. When both components are removed, the model's performance collapses entirely, producing distorted and ineffective results. These findings confirm that LBE and IBG are integral and complementary: LBE provides the semantic *what* to change, while IBG provides the spatial *where* to apply it.

## 5 EVALUATION FOR LATENT BASIS

Figure 7 shows the similarity values $a = S_{\mathcal{C}}(V^r, V^c)$ between latent bases at different denoising timesteps for two editing directions ("Beard" and "Big Lip"). We observe that similarity is higher at larger timesteps and decreases as denoising progresses. This behavior validates several key properties of our method:

**Linearity across timesteps.** The smooth, continuous decrease in similarity suggests that the local linearity assumption holds consistently throughout the denoising process. Sharp discontinuities would indicate breakdown of linearity, but the gradual transition demonstrates stable geometric structure in the latent space. At early timesteps (high noise), the latent bases $V^r$ and $V^c$ are more similar because noise dominates the latent space, making the linear approximation particularly valid. As denoising progresses, the bases diverge smoothly, indicating that the linear region accommodates the growing semantic differences between reconstruction and editing paths.

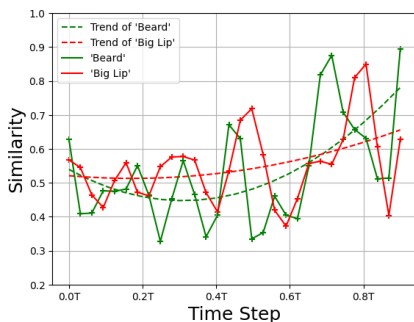

Figure 6: The similarity between the latent basis of the original video and the manipulated video evolves as the denoising progresses.

**Adaptive guidance mechanism.** The varying similarity across timesteps demonstrates why our adaptive threshold mechanism (using $1 - a$ quantiles) is crucial. At early stages where similarity is high ($a \approx 0.8$-$0.9$), our method applies minimal editing, preserving the coarse structure. At later stages where similarity drops ($a \approx 0.4$-$0.5$), larger editing regions are permitted, allowing fine-grained attribute manipulation. This adaptive behavior provides robustness even if the linearity assumption weakens at certain timesteps.

## 6 CONCLUSION

In this work, we introduced FlowGuide, a unified framework for high-fidelity face editing in both images and videos. We addressed the key challenge of disentangling identity from editable attributes by leveraging the geometric properties of the diffusion model's latent space. Our approach treats semantic attributes as linear subspaces and uses a novel guidance mechanism, consisting of Latent Basis Extraction (LBE) and Implicit Basis Guidance (IBG), to steer the generation process. By operating on the geometric alignment of these subspaces, our method confines edits to target attributes while preserving identity and temporal coherence. Extensive experiments demonstrate that FlowGuide achieves state-of-the-art performance, striking a superior balance between edit fidelity and attribute modification. Our work opens a promising direction for more controllable and geometrically-grounded manipulation in generative models.

## 7 LIMITATION DISCUSSION

While our method achieves state-of-the-art performance, several limitations remain. First, operating in the diffusion model's latent space can lead to over-smoothing in high-motion scenarios and unrealistic blending when adding hard-edge accessories like sunglasses. This represents a fundamental trade-off, we achieve superior identity preservation but at the cost of some visual artifacts. Second, perfect attribute disentanglement remains elusive; correlated features in training data lead to minor unintended changes of edits. Finally, our method inherits the limitations of the underlying diffusion model, restricting edits to what the latent space can represent and requiring fine-tuning for optimal performance on new domains.

## 8 ACKNOWLEDGMENT

The research was supported in part by Theme-based Research Scheme under Grant T45-205/21-N from Hong Kong RGC, and Generative AI Research and Development Centre from InnoHK.

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

## A    USE OF LLMs

We used LLMs for tasks such as improving grammar, refining phrasing, and ensuring consistency in language. The core ideas, experimental design, and interpretation of results are solely the work of the authors. All final content was reviewed and edited by the authors to ensure its accuracy and originality.

## B    RELATED WORK

### B.1    LATENT SPACE ANALYSIS

The study of latent spaces has gained significant attention in recent years. In the field of Generative Adversarial Networks (GANs), researchers have proposed various methods to manipulate the latent space to achieve the desired effect in the generated images (Ramesh et al., 2018; Patashnik et al., 2021; Abdal et al., 2021; Shen & Zhou, 2021; Härkönen et al., 2020). More recently, several studies have examined the geometrical properties of latent space in GANs and utilized these findings for image manipulations (Choi et al., 2021; Zhu et al., 2021). Some studies have applied Riemannian geometry to analyze the latent spaces of deep generative models (Arvanitidis et al., 2017; 2020; Chen et al., 2018; Lee & Park, 2023; Lee et al., 2022; Shao et al., 2018). (Shao et al., 2018) proposed a pullback metric on the latent space from image space Euclidean metric to analyze the latent space's geometry. This method has been widely used in VAEs and GANs because it only requires a differentiable map from latent space to image space. And (Park et al., 2023) extend it into diffuison models (DMs) to investigate the geometry of latent space of DMs to facilitate the image editing. However, it is challenging for the pullback metric to accurately capture the geometry of the latent space from the image space, as the image space contains excessive information, making it difficult to identify the correct directions for editing.

## C    EVALUATION FOR LATENT BASIS

By calculating the similarity between the latent basis of the original video and the manipulated video under a specific editing direction, we can quantify the degree of change introduced during editing. This similarity metric serves as a guide for the editing process, enabling more precise adjustments and ultimately improving the overall quality of the edits. In Figure 7, we present the change in similarity values at different denoising time steps for two editing directions: "Beard" and "Big Lip."

As observed, the similarity is higher at larger time steps and lower at smaller time steps. At larger time steps, increased noise in the latent space causes the original and edited videos to share similar latent bases. Conversely, at smaller time steps, reduced noise allows the latent basis to better capture encoding features, creating greater distinction between original and edited content.

Furthermore, this observation aligns with the understanding that the model initially focuses on low-frequency signals during the early stages of the generative process, where the similarities between the original and edited videos are more pronounced. Over time, the model progressively shifts its attention to high-frequency signals, which highlight the introduced target attribute and the differences between the two videos. This result reinforces the common view of the coarse-to-fine behavior exhibited by diffusion models throughout the generative process (Kim et al., 2023).

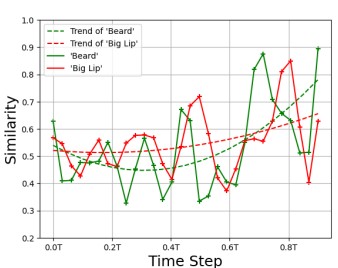

Figure 7: The similarity between the latent basis of the original video and the manipulated video evolves as the denoising progresses.

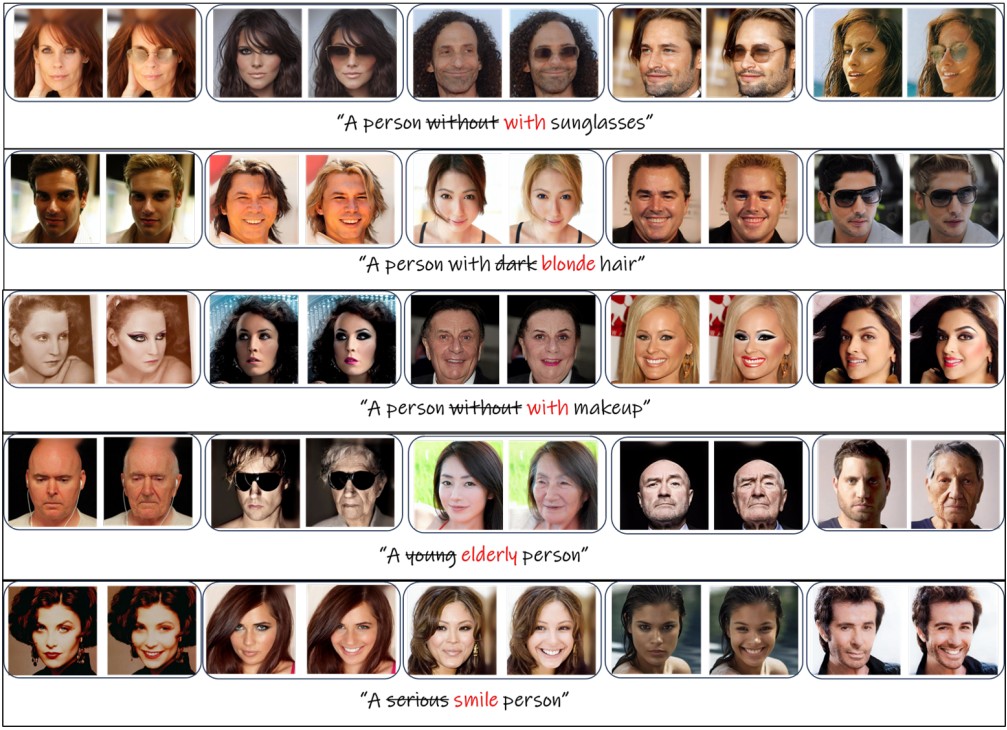

Figure 8: The editing sample visualization of our methods with different editing instructions.

## D  DEEPFAKE DETECTION ANALYSIS

To assess the detectability of videos edited by our method and baseline approaches, we conduct experiments using the state-of-the-art deepfake detection model from FaceForensics++ Rossler et al. (2019). Understanding the detectability of edited content is important for responsible AI development and helps identify potential misuse scenarios.

Table 4: Deepfake detection rates using FaceForensics++ detector on both image and video editing methods. Higher detection rates indicate easier identification of manipulated content. Lower rates suggest more natural-looking edits.

| Image Editing Methods | | | |
|---|---|---|---|
| **Method** | **Type** | **Detection Rate (%)** | **Naturalness** |
| **Ours** | Diffusion | **78.0** | **Most Natural** |
| h-edit Nguyen et al. (2025) | Diffusion | 79.5 | High |
| PnP Ju et al. (2023) | Diffusion | 81.0 | Moderate |
| EF Huberman et al. (2024) | Diffusion | 82.1 | Low |
| Video Editing Methods | | | |
| **Method** | **Type** | **Detection Rate (%)** | **Naturalness** |
| **Ours** | Diffusion | **72.5** | **Most Natural** |
| STIT Tzaban et al. (2022) | GAN | 85.5 | High |
| DVA Kim et al. (2023) | Diffusion | 91.5 | Moderate |
| Latent-trans Yao et al. (2021) | Transformer | 99.5 | Low |

As shown in Table 4, our method consistently achieves the lowest detection rates in both image and video editing tasks. For image editing, our method achieves a detection rate of 78.0%, outperforming h-edit (79.5%), PnP (81.0%), and EF (82.1%). For video editing, our method achieves 72.5%, significantly lower than baseline methods (85.5%-99.5%).

These results suggest that our edited content produces more natural-looking results that are harder to detect as manipulated media. We hypothesize that this is due to: (1) our method's superior preservation of temporal consistency in videos and natural facial dynamics throughout the sequence, and (2) better semantic coherence in edited regions that maintains the statistical distribution of real content. The lower detection rates in video editing (72.5%) compared to image editing (78.0%) demonstrate that our temporal modeling provides additional naturalness that is harder for detectors to identify.

## E   FACE IMAGE EDITING EXPERIMENT

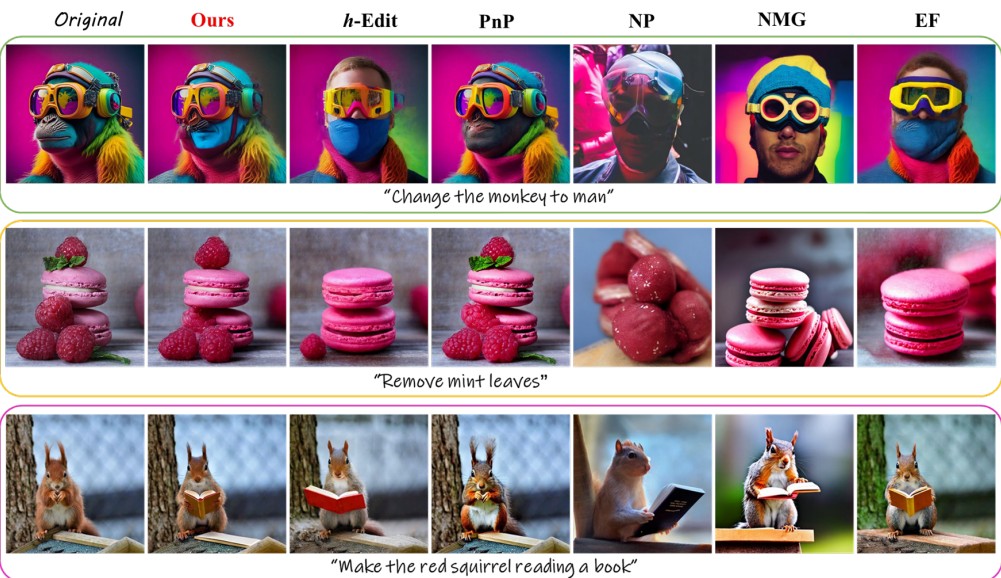

Figure 9: The editing results of different method on the task "*change object*" (the first row and the second row) and "*add object*" (the third row).

### E.1   EXPERIMENT SETTING

**Implementation Details:** For all face image editing experiments, we use the pre-trained Stable Diffusion v1.4 as our base model. We employ DDIM inversion for the encoding process, with both the inversion and denoising sampling steps set to 50.

A key aspect of our method is the use of distinct conditions for the inversion and denoising stages. For the initial DDIM inversion, both the reconstruction and editing paths use the same condition, $c_{org} = c_{edit}$, which corresponds to the source prompt (e.g., "a person with long hair") as defined in Appendix E.3. This ensures that both processes start from an identical noisy latent, $X_T$.

During the denoising phase, the conditions diverge to enable guided editing. The reconstruction path uses the original condition $C^r$ (derived from the source prompt), while the editing path uses the target condition $C^c$ (derived from the target prompt, e.g., "a person with short hair"). This setup allows our guidance mechanism to measure and control the semantic changes between the two paths. All experiments were conducted on a single NVIDIA RTX 4090 GPU.

**Implementation of Baseline Methods:** We use the following baseline methods for face image editing tasks, and all the baseline methods are implemented using the same base model and inversion process, the hyper-parameters are set to the same as the original paper:

- **h-Edit (Nguyen et al., 2025):** A hierarchical editing framework that decomposes the editing process into multiple semantic levels for more granular control over different attributes.

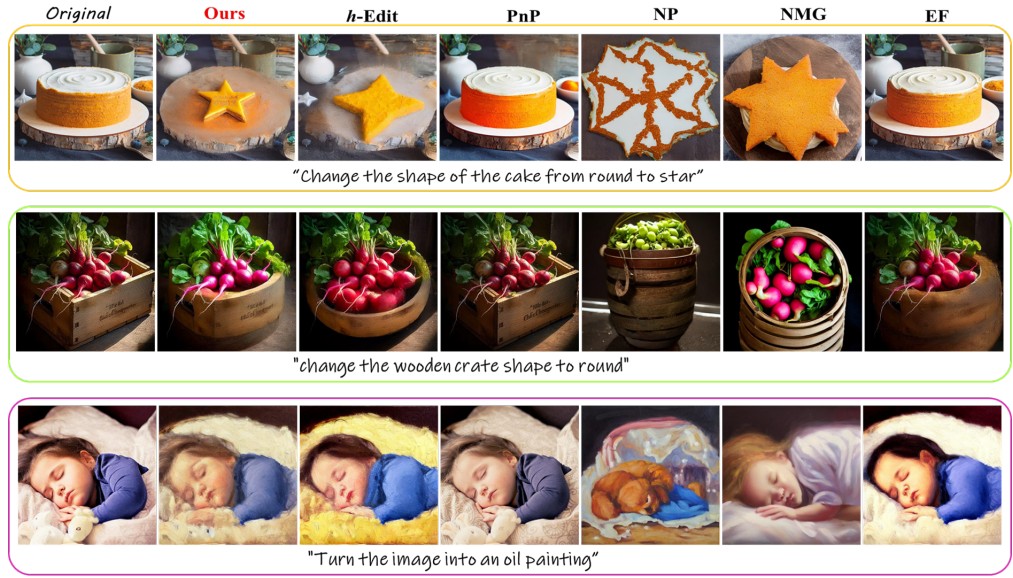

Figure 10: The editing results of different method on the task "*change attribute content*" (the first row), "*change attribute pose*" (the second row) and "*change style*" (the third row).

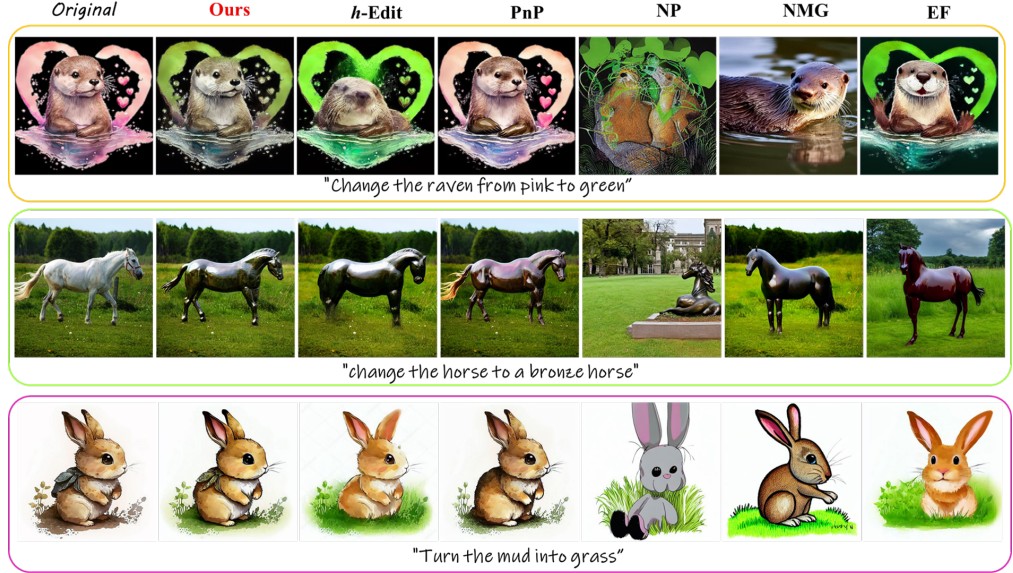

Figure 11: The editing results of different method on the task "*change attribute color*" (the first row), "*change attribute material*" (the second row) and "*change background*" (the third row).

- **PnP Inversion (Ju et al., 2023):** A plug-and-play method that avoids costly optimization by injecting features from the original input directly into the denoising process to guide the generation.

- **Noise Map Guidance (NMG) (Cho et al., 2024):** Leverages the structure of noise maps from the inversion process to guide the denoising steps, aiming to better preserve fine details and image structure.

- **Negative Prompt Inversion (NPI) (Miyake et al., 2023):** An efficient optimization-free method that uses the original text prompt embedding to approximate the null-text embedding, speeding up the inversion process.

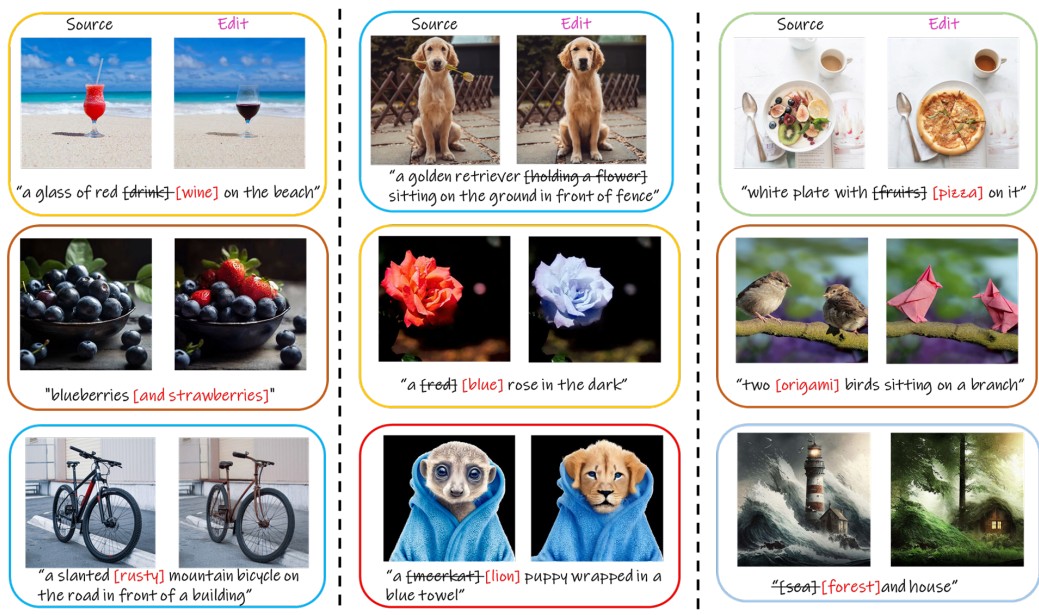

Figure 12: The editing results of our method on PIE benchmark.

- **Edit Friendly (EF) (Huberman et al., 2024):** A technique that employs random inversion rather than deterministic inversion, achieving high-quality reconstruction without needing attention map modifications.

### E.2 INVERSION PROCESS FOR IMAGE EDITING

We use the DDIM inversion process to invert the original image into the latent space for face image editing tasks. DDIM inversion is a deterministic process designed to find a noise latent $X_T$ that, when used as the starting point for the standard DDIM denoising process, reconstructs the original input image $X_0$ with high fidelity. This allows for the manipulation of real images by first inverting them into the latent space and then denoising them with a modified text prompt or condition.

The inversion process is iterative, progressively adding noise to the input image $X_0$ over $T$ timesteps. Starting with $X_0$ (the clean image), for each step $t$ from 1 to $T$, we calculate the next latent $X_t$ based on the previous latent $X_{t-1}$. The core of the process relies on using the pre-trained noise prediction network $\epsilon_\theta$ to estimate the noise that would have been present at step $t-1$, and then using this estimate to project forward to step $t$.

The update rule for each step of the DDIM inversion is as follows:

$$X_t = \sqrt{\frac{\bar{\alpha}_t}{\bar{\alpha}_{t-1}}} X_{t-1} + \left( \sqrt{1 - \bar{\alpha}_t} - \sqrt{\frac{\bar{\alpha}_t(1 - \bar{\alpha}_{t-1})}{\bar{\alpha}_{t-1}}} \right) \cdot \epsilon_\theta(X_{t-1}, t-1, c_{org}) \tag{7}$$

where:

- $X_{t-1}$ is the latent from the previous step (with $X_0$ being the initial image).
- $\bar{\alpha}_t = \prod_{i=1}^{t} \alpha_i$ is the cumulative product of the noise schedule coefficients $\alpha_i = 1 - \beta_i$.
- $\epsilon_\theta(X_{t-1}, t-1, c_{org})$ is the noise predicted by the UNet model for the latent $X_{t-1}$ at timestep $t-1$, under the original condition $c_{org}$.

By iteratively applying this equation from $t = 1$ to $T$, we obtain a trajectory of latents $\{X_1, X_2, \ldots, X_T\}$. The final latent, $X_T^r$, serves as the encoded representation of the original image. For editing, this latent is then used as the starting point for the denoising process, but guided by a new target condition $c_{edit}$ to generate the manipulated image.

### E.3 Constructing Face Image Editing Prompts

To systematically evaluate the performance of our face image editing framework, we constructed a standardized set of editing prompts. Each prompt is designed to test a specific, common facial attribute modification. The construction process for each data point follows a consistent structure, including a source prompt that describes the original image and a target prompt that describes the desired edited outcome.

For each editing task, we define the following components:

- **Source Prompt:** A brief textual description of the initial state of the attribute in the source image (e.g., "a person without sunglasses," "a person with long hair"). This prompt is used to generate the original condition, $c_{org}$.

- **Target Prompt:** A corresponding textual description of the desired state of the attribute after editing (e.g., "a person with sunglasses," "a person with short hair"). This prompt is used to generate the target condition, $c_{edit}$.

- **Editing Instruction:** A clear, human-readable instruction that specifies the transformation to be performed (e.g., "Add sunglasses to the person's face," "Change the hair length from long to short").

We curated a diverse set of common face editing tasks to ensure comprehensive evaluation. The primary editing axes we considered include:

- **Accessories:** Adding or removing items like sunglasses.
- **Hairstyle:** Modifying hair length or color (e.g., long to short, dark to blonde).
- **Age:** Changing the perceived age of the person (e.g., young to elderly).
- **Makeup:** Applying or removing makeup.
- **Expression:** Altering facial expressions (e.g., serious to smiling).

This structured approach to prompt construction allows for consistent and reproducible experiments, ensuring that all baseline methods are evaluated under the same conditions and that the performance of our model can be fairly assessed across a range of common and important face editing scenarios.

### E.4 Additional Visualization of Face image editing

We provide additional visualization of face image editing results in Figure 8. It can be seen that our method can successfully edit the face image, and the editing results are natural and realistic.

### E.5 PIE Benchmark Results

#### E.5.1 Main Results

The quantitative results on the PIE benchmark, summarized in Table 5, highlight the efficacy of our proposed method. A key observation across the baselines is the inherent trade-off between edit conformance and fidelity to the original image. For instance, methods such as PnP Inversion demonstrate strong performance in consistency metrics (DINO Dist, MSE Dist), indicating minimal deviation from the source image, but this comes at the cost of lower alignment with the target prompt (CLIP Sim). Conversely, methods like EF and NMG achieve high CLIP similarity by making more aggressive edits, which compromises image quality (PSNR, LPIPS) and consistency.

In contrast, our FlowGuide strikes a more effective balance across these competing objectives. It achieves the second-best performance on average across all quality and consistency metrics, surpassed only by the highly conservative PnP Inversion, while simultaneously maintaining a competitive CLIP Similarity score. This suggests that our geometrically-grounded guidance mechanism is not merely preserving the original image, but is enabling precise, targeted edits. By confining modifications to the intended semantic regions, our method preserves the overall fidelity and structure of the source image without sacrificing the desired semantic change, thereby achieving a superior position on the fidelity-conformance trade-off curve.

Table 5: The text-guided image editing performance of different editing methods.

| Method | Edited Image Quality | | | Edited Performance | | Consistency | |
|---|---|---|---|---|---|---|---|
| | PSNR ($\uparrow$) | LPIPS ($\downarrow$) | SSIM ($\uparrow$) | CLIP Sim ($\uparrow$) | Local CLIP ($\uparrow$) | DINO Dist ($\downarrow$) | MSE Dist ($\downarrow$) |
| EF Huberman et al. (2024) | 17.624 | 0.1771 | 0.7306 | **27.127** | 0.1520 | 0.0661 | 0.0229 |
| NMG Cho et al. (2024) | 14.075 | 0.3189 | 0.6063 | 27.053 | 0.1563 | 0.1257 | 0.0492 |
| NP Miyake et al. (2025) | 14.510 | 0.3262 | 0.6028 | 24.182 | **0.1968** | 0.1266 | 0.0446 |
| h-Edit Nguyen et al. (2025) | 19.657 | 0.1397 | 0.7711 | 26.815 | 0.1878 | 0.0536 | 0.0156 |
| FlowGuide | **22.021** | **0.1006** | **0.8047** | 26.422 | 0.1754 | **0.0349** | **0.0091** |

### E.5.2 VISUALIZATION

**Comparison Results Visualization:** We present the comparison of editing results of different method on the all eight tasks in PIE benchmark. As shown in Figure 9, Figure 10 and Figure 11, our method achieves the best performance on all the tasks. In Figure 9, "change object" and "add object" are included. In Figure 10, "change attribute content", "change attribute pose" and "change style" are included. In Figure 11, "change attribute color", "change attribute material" and "change background" are included. We can see that our method can successfully edit the object in the image, and the editing results are more natural and realistic than the baseline methods.

**Visualization of Our Method:** We provide additional visualization results of our method on PIE benchmark. As shown in Figure 12, our method can successfully edit the object in the image, and the editing results are more natural and realistic than the baseline methods.

## F FACE VIDEO EDITING EXPERIMENT

### F.1 INVERSION FOR VIDEO EDITING

To encode the conditions related to the target attribute into the video, we first obtain the embedding for the original frames using a pre-trained condition generator, denoted as $\mathcal{E}_c$: $c_{org} = \mathcal{E}_c(X)$. Next, we utilize a pre-trained encoder $\mathcal{E}_e$ to jointly encode the video frames and the associated embedding into conditions (the process of obtaining $c_{edit}$ can refer to Appendix F.4), which are then used as conditions during the denoising process:

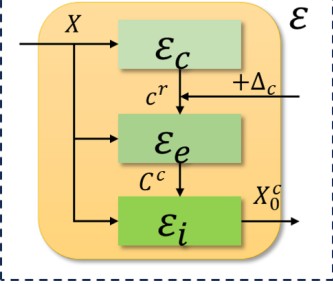

Figure 13: The architecture of encoder $\mathcal{E}$, consists of $\mathcal{E}_c$, $\mathcal{E}_e$ and $\mathcal{E}_i$.

$$\mathcal{C}^r = \mathcal{E}_e(X, c_{org}), \ \mathcal{C}^c = \mathcal{E}_e(X, c_{org} + c_{edit}) \tag{8}$$

where $\mathcal{C}^r$ and $\mathcal{C}^c$ are utilized as conditions for the denoising of the original and manipulated frames, respectively. And the input representations at time step $t = 0$ are derived using a frozen input encoder $\mathcal{E}_i$: $X_0^r = \mathcal{E}_i(X, \mathcal{C}^r)$ and $X_0^c = \mathcal{E}_i(X, \mathcal{C}^c)$, $X_0^r$ represents the original input representation and $X_0^c$ serves as the conditional input representation for manipulation.

After obtaining the encoded input representations $X_0^r$, $X_0^c$, the forward diffusion can be applied:

$$q(X_t^r | X_0^r) = \mathcal{N}(X_t^r; \sqrt{\alpha_t} X_0^r, (1 - \alpha_t)\epsilon_t^r), \ \epsilon_t^r = \mathcal{F}_\theta(X_0^r, t, \mathcal{C}^r) \tag{9}$$

where $\mathcal{F}_\theta$ denotes a pre-trained noise estimator, and $X_t^r$ represents the noisy representation at diffusion step $t$. The parameter $\alpha_t$ controls the noise scale at step $t$. Through this process, $X_T^r$ is generated by the forward diffusion process. Similarly, the forward diffusion process is applied to $X_0^c$ to obtain $X_T^c$.

### F.2 EXPERIMENT SETTINGS

**Implementation details.** FlowGuide uses a diffusion autoencoder with a UNet as the noise estimator. To enhance the model's ability to reconstruct the background in face videos, we fine-tune the pre-trained diffusion autoencoder from (Kim et al., 2023) on the HDTF dataset (the details of fine-tuning the diffusion autoencoder can refer to Appendix F.5). Note that during the editing process, the pre-trained diffusion autoencoder model remains frozen. We use the DDIM sampler, setting the the reverse time step and the inference time step to 50. The batch size for inference is set to 1, and all

inference is performed on 4 RTX4090 GPUs. For face video editing, we didn't use the edit friendly guidance, the consistency is realized solely by our method solely. We report the inference time of our method and the baseline methods in Appendix F.6.

**Implementation of Baselines.** We select several state-of-the-art methods for comparison: the diffusion-based editing method DVA Kim et al. (2023) and the transformer-based method Latent-trans Yao et al. (2021). For GAN-based methods, we include STIT Tzaban et al. (2022), TCSVE Xu et al. (2022), PTI Roich et al. (2022), and StyleCLIP Patashnik et al. (2021).

It is important to emphasize that, for a fair evaluation of reconstruction capabilities, all methods only use the original videos as input. None of the methods have access to the original videos during the output generation phase, ensuring that the reconstruction quality reflects the true performance of each editing approach without reliance on the input data.

- DVA Kim et al. (2023): For the implementation of DVA, we use their CLIP-based editing method, and the editing scale $\alpha$ is set to 0.25 as recommended in their paper, and the input texts of the CLIP-based editing method are "Face" and "Face with *" for original video and the target manipulated video, other experiment settings are used the default settings.

- TSCVE Xu et al. (2022) We use the default settings as recommended, and the frames of the videos are resized to 512. We also use the output frames directly, without blending them into the original video frames.

- Latent-trans Yao et al. (2021): For the implementation of Latent-trans, we set the scaling factor $\alpha$ as 1.5 and the other settings are kept as recommended. And we use the output frames directly, the output frames are not blended with the original input frames.

- STIT Tzaban et al. (2022): We run edits with stitching tuning, and the edit ranges is set to 10101, the parameter $\beta$ is set to 0.2 and the *outer_mask_dilation* is set to 50. Other settings are kept as recommended. The output frames are used directly as well.

- PTI Roich et al. (2022): We use the default settings as recommended, and the frames of the videos are resized to 1024. We also use the output frames directly, without blending them into the original video frames.

- StyleCLIP Patashnik et al. (2021): We train the mappers of input videos with the default settinfs and use the attributes as the descriptions. Then we use the default settings to edit the videos and the output frames are used directly.

### F.3 RECONSTRUCTION EVALUATION

For video editing tasks, it is essential that the model can accurately reconstruct the original video from its encoded representation. To achieve this, we fine-tune the pre-trained diffusion autoencoder to enhance its ability to accurately reconstruct both the background and human face. We evaluate the reconstruction performance of FlowGuide against all baseline methods on the HDTF and VoxCeleb datasets, with the results reported in Table 6.

Table 6: The reconstruction performance of our FlowGuide and baselines on HDTF and Voxceleb datasets. The reported values are the mean of the averaged per-frame measurements for each video.

| Method | HDTF | | | | VoxCeleb | | | |
|--------|------|------|------|------|----------|------|------|------|
| | SSIM ($\uparrow$) | LPIPS ($\downarrow$) | MSE ($\downarrow$) | FID ($\downarrow$) | SSIM ($\uparrow$) | LPIPS ($\downarrow$) | MSE ($\downarrow$) | FID ($\downarrow$) |
| StyleCLIP | 0.6653 | 0.1984 | 0.0125 | 136.52 | 0.4830 | 0.3028 | 0.0183 | 233.60 |
| STIT | 0.5202 | 0.3978 | 0.0617 | 244.60 | 0.6669 | 0.2769 | 0.0513 | 179.27 |
| PTI | 0.6347 | 0.2476 | 0.0256 | 168.12 | 0.4737 | 0.3434 | 0.0337 | 227.43 |
| Latent-trans | 0.7035 | 0.1571 | 0.0068 | 137.70 | 0.6017 | 0.2208 | 0.0076 | 217.96 |
| DVA | 0.9448 | 0.0584 | 0.0003 | 33.531 | 0.9696 | 0.0130 | 0.0006 | 44.458 |
| FlowGuide | **0.9715** | **0.0108** | **0.0001** | **23.432** | **0.9779** | **0.0095** | **0.0004** | **24.840** |

Table 6 clearly demonstrates that our method achieves significantly better reconstruction performance compared to baseline methods on both the HDTF and VoxCeleb datasets. This highlights the superior ability of our model to faithfully reconstruct fine details in both the background and

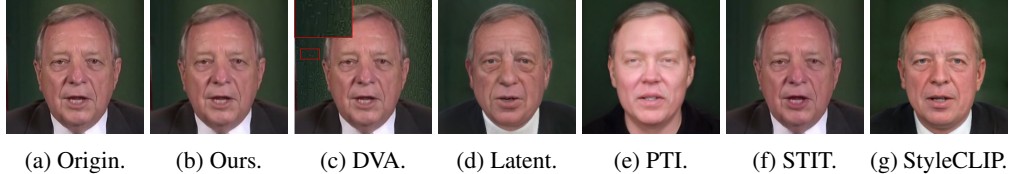

(a) Origin.    (b) Ours.    (c) DVA.    (d) Latent.    (e) PTI.    (f) STIT.    (g) StyleCLIP.

Figure 14: The comparison of the images reconstructed by our FlowGuide and five baseline methods with the original input image.

human face, underscoring its robustness and generalizability. We further provide a visual comparison of the reconstruction performance across different methods in Figure 14. It can be seen from Figure 14 that baseline methods struggle to either preserve the identity of the characters or retain the background features. In contrast, our FlowGuide shows clear superiority in reconstructing the face videos, delivering more accurate restoration of both facial identity and background details. This enhanced reconstruction ability makes FlowGuide particularly effective for tasks where maintaining consistency between the original content and the edited results is crucial, highlighting its robustness in video manipulation.

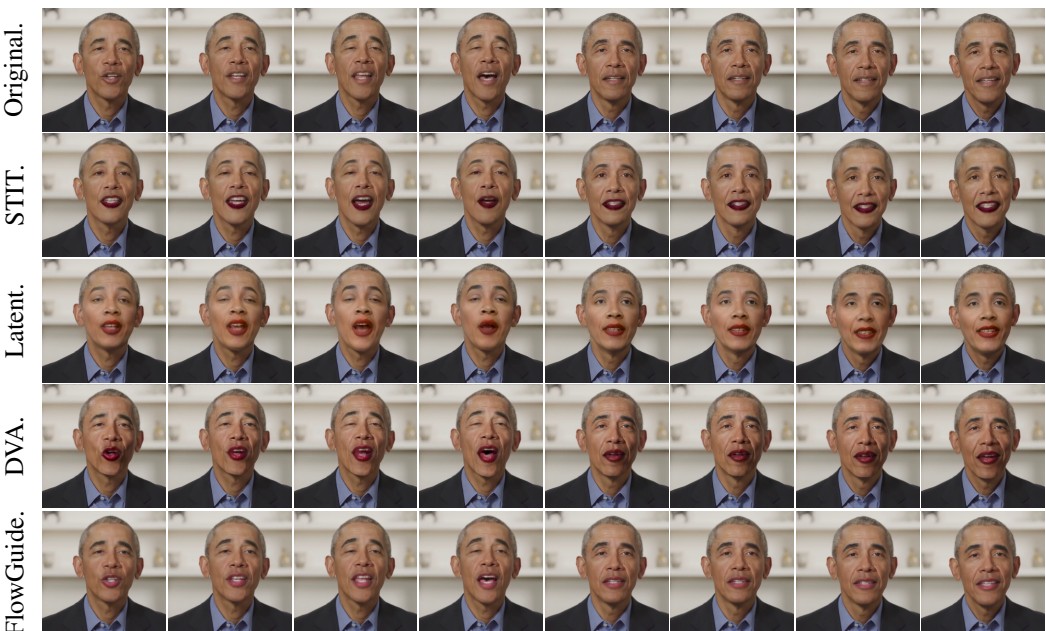

Figure 15: Comparison of editing performance of our FlowGuide to the previous video editing methods for editing direction 'Libstick'.

## F.4    OBTAIN CONDITION

To edit videos using diffusion-based models, the editing directions must first be mapped into conditions. We achieve this by leveraging the pre-trained CLIP model Radford et al. (2021) to encode the editing directions. We denote original condition as $\mathcal{C}^r$ (see Equation 8), and represent the input with this original condition as $X_0^r$. The forward diffusion process is then applied to $X_0^r$ over the diffusion steps $\hat{T}$.

Next, the target conditions are initialized as $\hat{\mathcal{C}}^c = \mathcal{C}^r$. These target conditions are iteratively updated until the final conditions are obtained. At each diffusion step $t \in \hat{T}$, we compute the input $\hat{X}_t^c$ using the equation $\hat{X}_t^c = \mathcal{E}_i(X_t^0, \hat{\mathcal{C}}^c)$, ensuring that the editing directions are accurately incorporated into the denoising process.

The source text for $X_0^r$ is "face," and the target text is "face with $\delta$," where $\delta$ represents the target attribute. We use $I$ to denote the source text and $I_\delta$ to denote the target text. To quantify the differ-

ence between the source and target conditions, we utilize the CLIP loss function $\mathcal{L}_{clip}$ from Radford et al. (2021) to compute the loss. The loss function is formulated as:

$$\mathcal{L}_1 = \sum_{t=0}^{\hat{T}} \mathcal{L}_{clip}(I, X_t^r, I_\delta, \hat{X}_t^c) \tag{10}$$

This loss helps guide the model toward generating video frames that align with the target attributes defined by $\delta$.

Then to keep the consistency of the background information of the reconstructed frames under the target conditions with the original video frames, another loss function is used:

$$\mathcal{L}_2 = \frac{1}{\hat{T}} \sum_{t=0}^{\hat{T}} (X_t^r, \hat{X}_t^c) \tag{11}$$

and to control the updated conditions don't vary too much:

$$\mathcal{L}_3 = 1 - \frac{\mathcal{C}^r \hat{\mathcal{C}}^c}{||\mathcal{C}^r||||\hat{\mathcal{C}}^c||} \tag{12}$$

then the optimization object can be obtained as:

$$\mathcal{L} = w_1 \mathcal{L}_1 + w_2 \mathcal{L}_2 + w_3 \mathcal{L}_3 \tag{13}$$

where $w_1, w_2, w_3$ are constants. And through minimizing $\mathcal{L}$ until convergence, we could get the trained conditions $c_{edit} = \mathcal{C}^r - \hat{\mathcal{C}}^c$.

**Settings for Obtaining Conditions**

In this paper, we use the pre-trained CLIP model, specifically the ViT-B/32 version. The weights $w_1, w_2, w_3$ are set to 5, 1, and 3, respectively, and the forward time step $\hat{T}$ is set to 5. The learning rate is set to 0.002, with a batch size of 1 during training. The number of updating steps is fixed at 1000.

### F.5 FINETUNE DIFFUSION AUTOENCODER

We finetune the pre-trained diffusion autoencoder from Kim et al. (2023) on the HDTF dataset. The loss function used for finetuning consists of two components. The first component is the standard DDIM (Denoising Diffusion Implicit Models) loss function, represented as:

$$\mathcal{L}_{ddim} = \mathbb{E}_{\epsilon_t \sim \mathcal{N}(0,I)} ||\epsilon_t^r - \epsilon_t||_1 \tag{14}$$

where $\epsilon_t^r$ can refer to Equation 9 and $\epsilon_t$ is the true noise, $t \in T$. This loss is minimized to ensure accurate denoising and reconstruction during the finetuning process.

To enhance the robustness of the model to noise, we sample images given the time step with two different noise realizations, denoted as $\epsilon_1$ and $\epsilon_2$, where $\epsilon_1, \epsilon_2 \sim \mathcal{N}(0,1)$. The sampled images are represented as $\hat{X}_t^1$ and $\hat{X}_t^2$.

The loss function for this sampling process can be formulated as follows:

$$\mathcal{L}_{reg} = \mathbf{E}_{\epsilon_1, \epsilon_2 \sim \mathcal{N}(0,1)} ||\hat{X}_t^1 - \hat{X}_t^2||_1 \tag{15}$$

This loss encourages the model to accurately predict the noise for both sampled images, thereby improving its robustness against variations in noise during the denoising process.

The final optimization objective for finetuning the diffusion autoencoder is $\mathcal{L} = \mathcal{L}_{ddim} - \mathcal{L}_{ddim}$

**Settings for Finetuning the Diffusion Autoencoder**

We finetune the diffusion model on HDTF dataset. The learning rate is set to 1e-4 and the dropout is set to 0.1, and we sample from each videos 16 frames during each training step. The batchsize is set to 16, the total training steps is set to 120000. And we set the seed to 0, the diffusion step $T = 1000$. The experiment is performed on 4 RTX4090 GPUs.

## F.6 COMPUTATIONAL EFFICIENCY

o demonstrate the efficiency of our proposed method, we compare the inference time of editing one frame with the baseline methods. The results are shown in Table 7. Since that the GANs based methods only need one forward pass to generate the video, we only compare the inference time of our method and STIT and DVA. It can be seen that our method is more efficient than the baseline methods.

Table 7: The inference time of our method and the baseline methods.

|  | STIT | DVA | FlowGuide |
|---|---|---|---|
| Infer Time | 12.0 sec | 62.4 sec | 4.94 sec |

## F.7 ADDITIONAL RESULTS

**Multiple Editing Direction** We provide more manipulation results of a single video across multiple editing directions in Figure 16 and Figure 17. Our approach excels at handling highly intricate background details and dynamic scenes that include substantial head movements and speech—scenarios that typically challenge existing state-of-the-art methods. Furthermore, our method adeptly retains the stylistic elements of the original video, ensuring that the edited output blends seamlessly with the untouched portions. This results in an exceptionally natural appearance, with virtually no visible traces of editing. The ability to maintain such coherence across different editing tasks underscores the robustness and adaptability of our approach.

## F.8 CROSS-SUBJECT EDITING

We claim that the obtained condition $c_{edit}$ can be used to edit the video of different subjects. To verify this claim, we further evaluate the cross-subject editing capabilities of our proposed method. As shown in Figure 18, we can edit the video of different subjects with the same condition $c_{edit}$.

## F.9 MORE COMPARISON RESULTS

We provide more visualization results of our method and the baseline methods with editing direction "Mustache" in Figure 19. It can be seen that our method can handle the complex background and dynamic scene, and the edited output can blend seamlessly with the untouched portions.

## F.10 NON PASTE-BACK VISUALIZATION

To evaluate the performance of our method and baselines more thoroughly, we provide the visualization results of our method and the baseline methods in Figure 21 with editing direction "Smile" without paste-back the editing results to the original video. It can be seen that our method can handle the complex background and dynamic scene, and the edited output can blend seamlessly with the untouched portions.

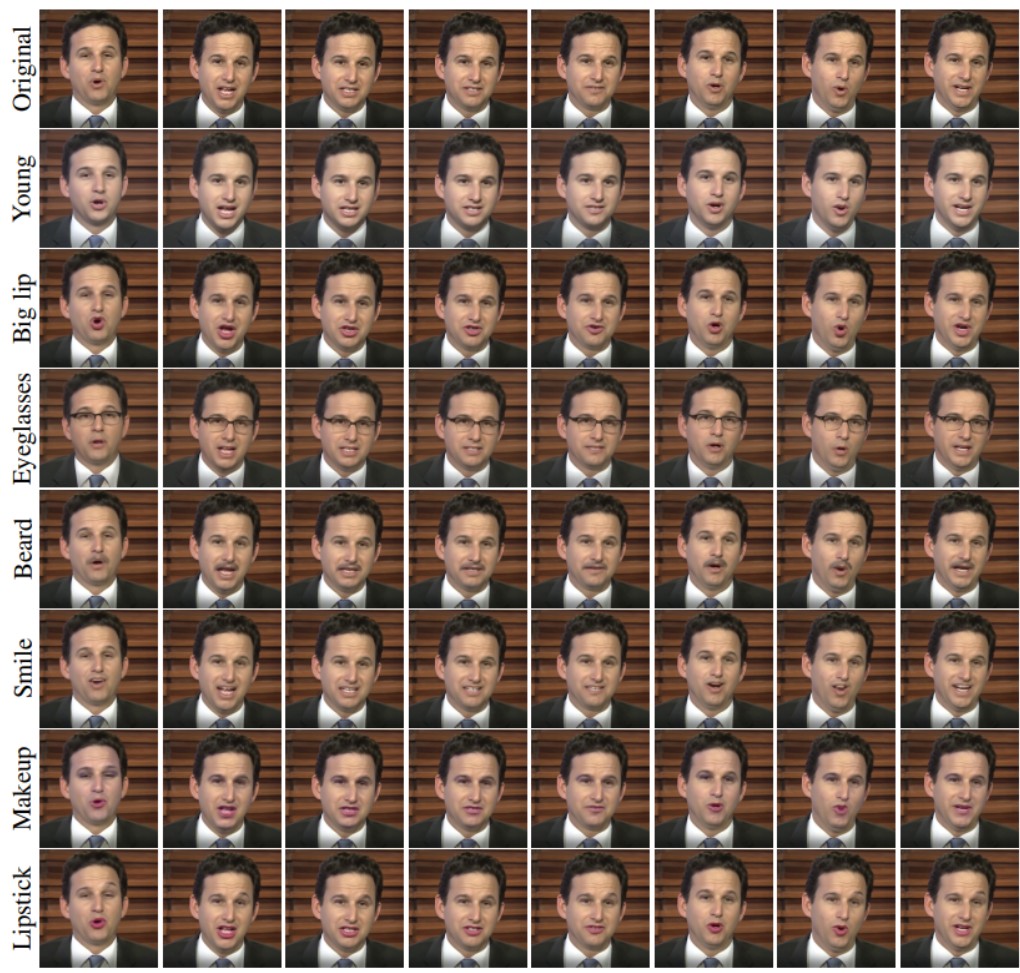

Figure 16: Manipulation results of our FlowGuide on a single video with different editing directions: "Beard" and "Big Lip", "Eyeglasses", "smile", "Young", "makeup" and "wearing Lipstick".

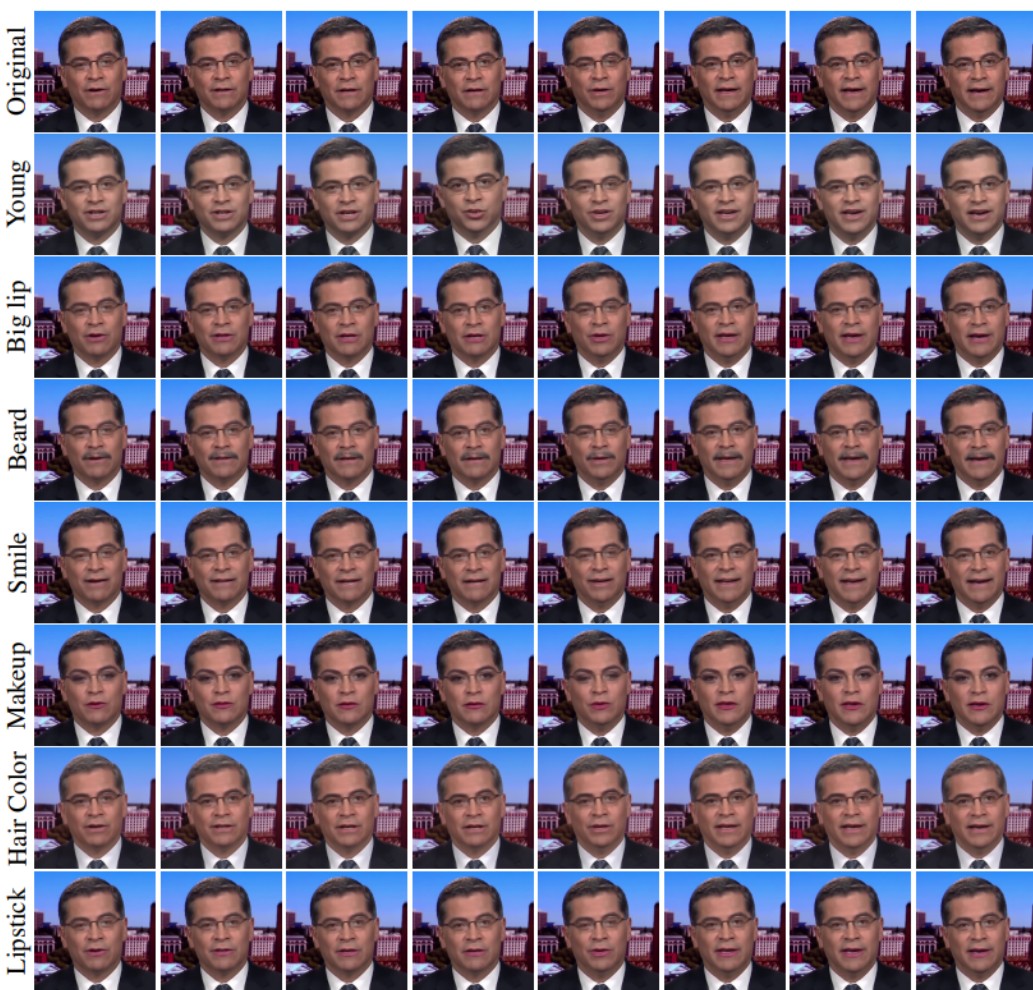

Figure 17: Manipulation results of our FlowGuide on a single video with two different editing directions: "Beard" and "Big Lip", "Hair Color", "smile", "Young", "makeup" and "wearing Lipstick"..

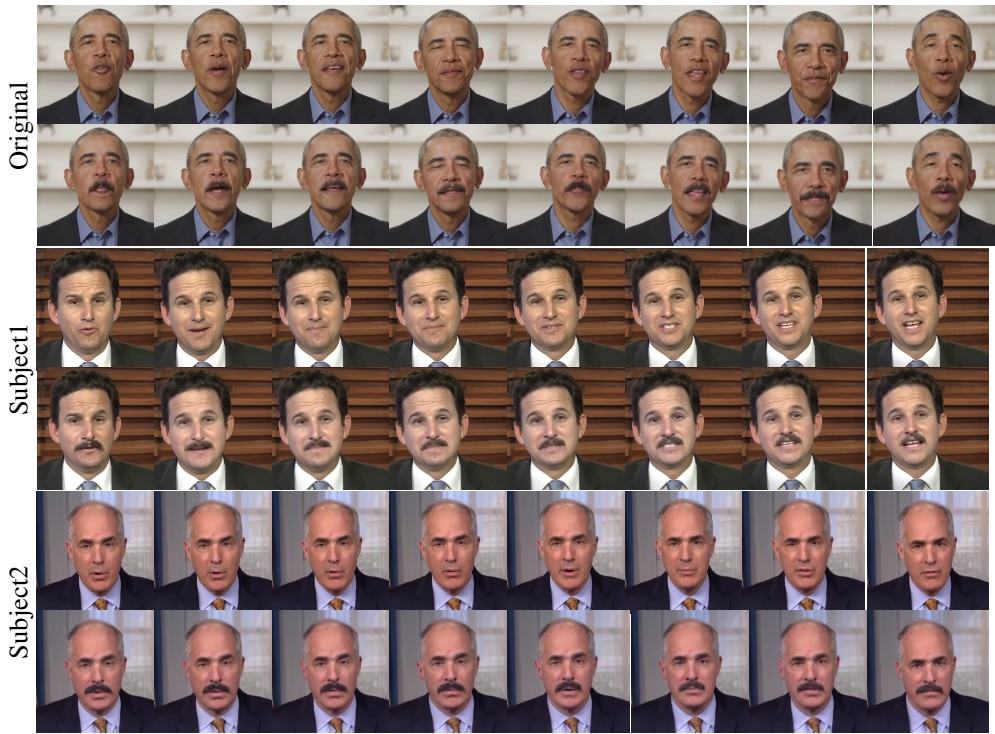

Figure 18: The cross-subject editing results of our method.

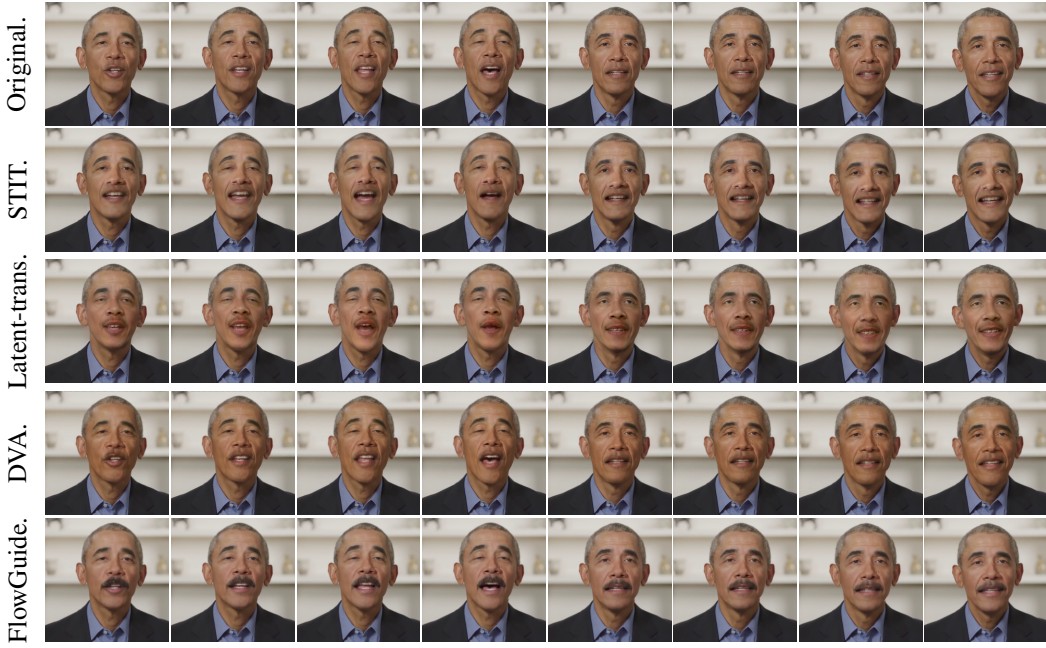

Figure 19: Comparison of editing performance of our FlowGuide to the previous video editing methods for editing direction 'Mustache'.

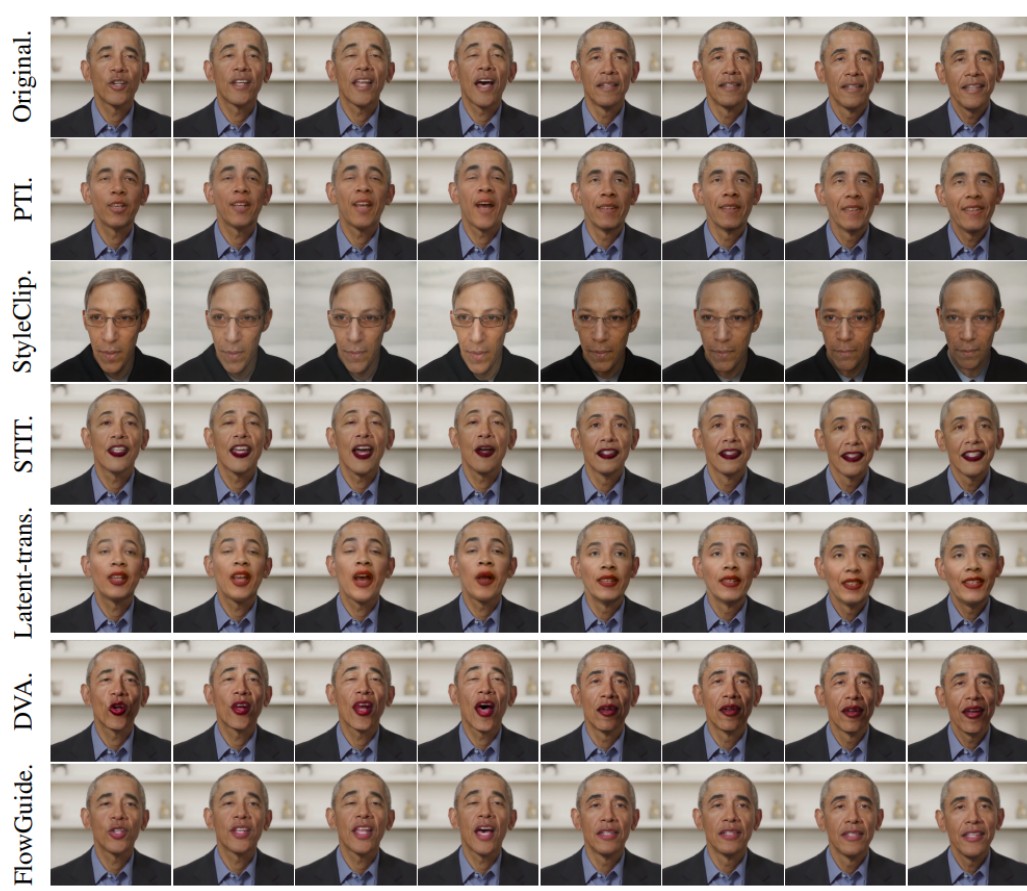

Figure 20: Comparison of editing performance of our FlowGuide to the previous video editing methods for editing direction 'Libstick'.

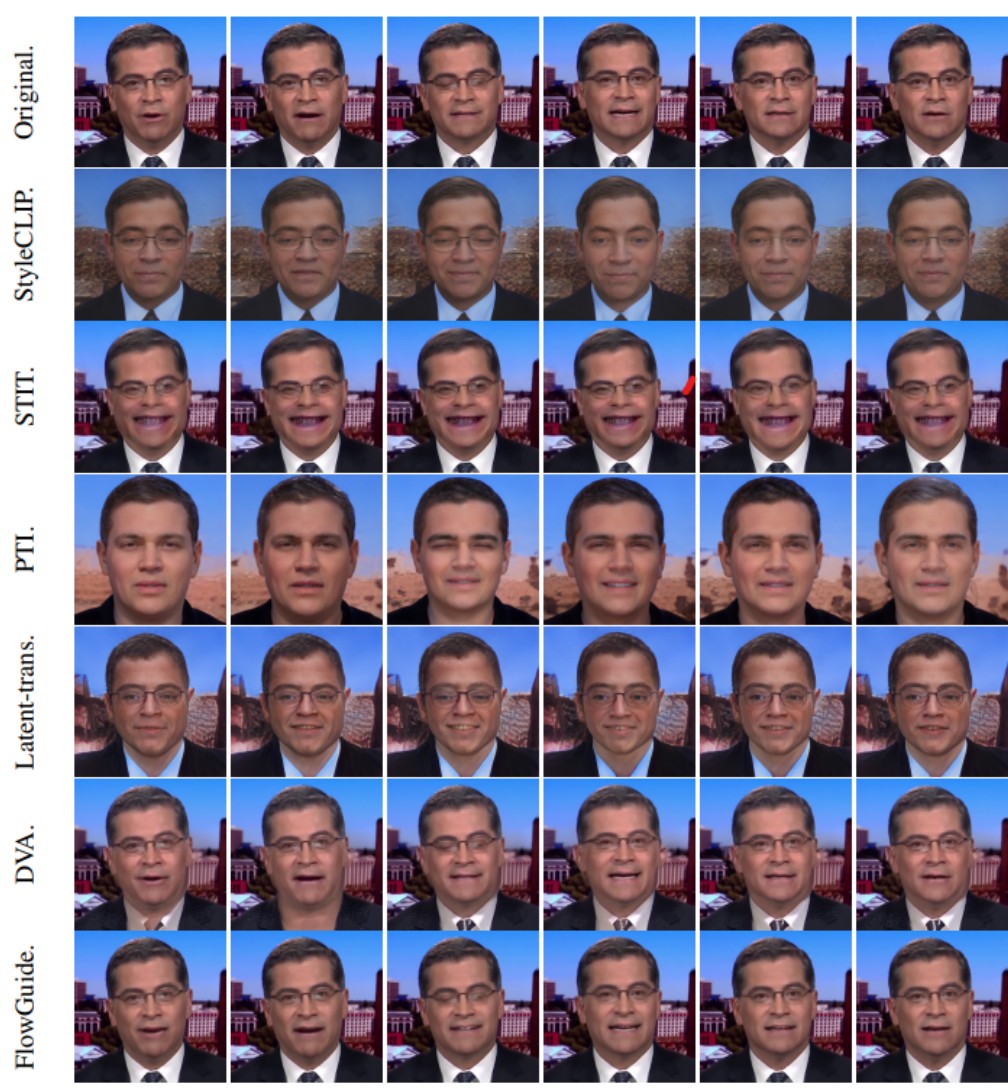

Figure 21: Comparison of editing performance of our FlowGuide to the previous video editing methods for editing direction 'Smile'.

