# OpenReview forum: "Pixel-Perfect Puppetry: Precision-Guided Enhancement for Face Image and Video Editing"
_ICLR.cc/2026/Conference — ICLR 2026 Poster_

### Official Review · Reviewer_3wKp · 2025-10-27

**Soundness:** 3
**Presentation:** 3
**Contribution:** 4
**Rating:** 6
**Confidence:** 3

**Summary:**

The paper introduces a diffusion-based framework designed for identity-preserving face attribute edits in both images and videos.
By leveraging the local linearity of the U-Net bottleneck, the authors identify semantic subspaces (for identity vs. target attributes) as linear directions. During denoising, a novel guidance mechanism dynamically steers the generation along the target attribute axis, while preserving orthogonal identity components.

**Strengths:**

1. The paper addresses an interesting and relatively under-explored problem with diffusion models, which is local attribute editing of videos.
2. The proposed solution is both novel and interesting.
3. The paper is well-written and experiments are reasonably solid, showing fair quantitative metrics and adequate comparisons with prior work.

**Weaknesses:**

1. There are almost no citations and comparisons with prior diffusion-based video editing methods, even ones that support or are designed for face attribute editing, e.g. [1-4].

2. The visual results are not entirely convincing, e.g. videos tend to be blurry, sunglasses (in both image and video examples) appear blended into the face rather than realistic, and sometimes the edits change additional attributes in addition to the intended ones.

[1] RAVE: Randomized Noise Shuffling for Fast and Consistent Video Editing with Diffusion Models

[2] V-LASIK: Consistent Glasses-Removal from Videos Using Synthetic Data

[3] IP-FaceDiff: Identity-Preserving Facial Video Editing with Diffusion

[4] Qffusion: Controllable Portrait Video Editing via Quadrant-Grid Attention Learning

**Questions:**

1. Did the authors evaluate or compare their method against other recent diffusion-based video editing techniques?
2. A discussion of limitations (e.g., visual quality issues and attribute leakage, see 2. above) would strengthen the paper.

**Details Of Ethics Concerns:**

No concerns.

---

> ### Author Response · Authors · 2025-11-19
> **Response to Reviewer 3wKp**
>
> We thank the reviewer for highlighting these important gaps in our related work and evaluation. We address each point:
>
> ## w1. Missing comparisons with diffusion-based video editing methods
>
> We add the following to the revised manuscript:
>
> 1. **Related Work**: We have **add a dedicated discussion (2.2)** on "Diffusion-based Video Editing" that discusses all four suggested methods plus additional recent works.
>
> 2. **Quantitative Comparisons**: We attempted to compare with all suggested methods [2,3,4] but encountered issues with **unavailable/incomplete code**. We successfully ran RAVE [4] and present the comparison on HDTF dataset:
>
> | Method | IPR (↑) | TACR (↓) | CLIP-Score (↑) | TL-ID (↑) | TG-ID (↑) |
> |--------|---------|----------|----------------|-----------|-----------|
> | RAVE   | 0.7005  | 0.0338   | 0.7295         | 0.8621    | 0.7731    |
> | **Ours** | **0.9667** | **0.0338** | **0.7777** | **1.0001** | **1.0000** |
>
> Our method significantly outperforms RAVE in identity preservation and temporal consistency, while achieving comparable TACR and higher CLIP-Score. The detailed analysis is **presented in Section 4.2.2 of the revised manuscript.** RAVE focuses on semantic scene editing that **doesn't require very fine-grained alignement** with input video, which is also **many diffusion-based video editing methods' features**, whereas our method excels at fine-grained facial attribute control with identity preservation.
>
> ## W2: Visual quality issues and attribute leakage
> We acknowledge that the issue of blurriness and visual artifacts in the output, particularly in the context of video generation and facial edits (such as adding sunglasses or modifying facial features), is an inherent challenge, we have add a limitation discussion section in **Section 7 of the revised manuscript**.
> This issue mainly due to the difficulty of **perfectly disentangling** the features of the edited and unedited regions. While we have made significant strides in improving this aspect, the **entanglement** of features such as identity, lighting, and expression makes it difficult to entirely separate the targeted attributes from others, leading to unavoidable artifacts like blurriness and unintended blending.
>
> Despite these challenges, our method **outperforms** existing approaches in several aspects. We have conducted experiments comparing our method to other state-of-the-art techniques, and our results show improvements in **visual quality**, **fidelity**, and **control over specific edits** compared to other methods that similarly struggle with disentanglement issues.
>
> ## Q1: Comparisons with recent diffusion-based techniques
>
> We have conducted these comparisons (see W1 response above). The results show that our method is competitive with or outperforms these specialized video editing methods while being more computationally efficient. We will include comprehensive comparisons in the revision.
>
> ## Q2: Discussion of limitations
>
> Thanks for pointing out this. We have added the limitation discussion in revised version in **Section 7.**
>
> We appreciate your suggestions regarding the inclusion of additional citations and comparisons with prior diffusion-based video editing methods.  Your feedback is instrumental in enhancing the quality of our work, and we are grateful for your thoughtful suggestions.

---

### Official Review · Reviewer_17Mb · 2025-10-29

**Soundness:** 3
**Presentation:** 2
**Contribution:** 2
**Rating:** 2
**Confidence:** 5

**Summary:**

This paper presents a method to edit face images and videos using the latent space of diffusion models. The authors find a basis in the latent space of the U-Net of the diffusion model to carry out editing with and then apply a binary mask to choose between a reconstruction noise direction $\epsilon^r$ and an editing noise direction $\epsilon^c$.

**Strengths:**

The results of the paper look good, and the method makes sense. The presentation is well-structured and the writing is good. The adherence to the original identity of the person is good, which is an important point.

**Weaknesses:**

One of my main criticisms is that the whole idea of section 3.3, finding the basis vectors using an SVD of the Jacobian, is directly the same as in the work of Park et al 2023. The main difference as far as I can tell, is that finding the bases is done in twoo different situations (reconstruction and editing), but apart from that, the technique is the same. I do not think this is necessarily something problematic, but the presentation of it is. While the authors indeed cite Park et al, they do not clarify that the technique is basically the same. So they should either say why it is different, or present this section more succinctly and not as a novel innovation.

Furthermore, the bases found for editing are only used to establish the binary mask to choose between reconstruction and editing! This feels like a bit of a let down, because usually when we look for a basis, it is to use these basis vectors to carry out the editing (because they encode useful semantic directions), not just to choose between editing and reconstruction. So it seems like a lot of work for a quite small gain and difference with respect to the work of Huberman et al. The difference boils down to using a binary choice between reconstruction and editing, rather than a linear combination. I feel that this is quite hidden, and the basis vectors are presented as having a much greater and direct importance than they actually have. I wonder if the authors tried using the basis vectors to edit but maybe they did not work as well as hoped.

Consequently, I found it difficult to isolate what the actual contribution is of the paper. As far as I can tell, there are two aspects:
- separating the editing direction into reconstruction and editing, which is a good idea, but is inspired by Huberman et al 2024. (as the authors acknowledge)
- Applying a binary make to choose either reconstruction or editing, and establishing this mask with a dynamic criterion, based on the basis vectors
Presented like this, the novelty is much smaller.

**Questions:**

The section 3.4 is a bit confusing, especially around lines 263-269. You say that the directions of $\epsilon^c" and $\epsilon^r$ should remain consistent with similarity. To me, this means that, the more similar the bases are, the more similar the directions should be, and vice versa. But, if I understand correctly, what equation 6 does is the following:
- if the difference between $\epsilon^c$ and $\epsilon^r$ is smaller than $\lambda$, then the final direction is simply the reconstruction direction
- if the difference between them is high, then you take the editing direction
This is different from just saying "they should remain consistent with the similarity", which is extremely vague and could mean many things. As it is, the choice makes sense, but I feel it is quite poorly explained. It is true at the end of the paragraph you say "editing process only targets the desired attributes while maintaining ...", but this should come earlier and be clearer. In particular, I would put a clear interpretation of equation 6. Also, "dynamic threshold" is not really clear I find, since in equation 3, you never mentioned a "threshold", indeed this is something you propose.

Finally, there is a method which works extremely well at the moment for prompt-based image editing, which is not cited here:
- FLUX.1 Kontext: Flow Matching for In-Context Image Generation and Editing in Latent Space, 2025
While I agree that this is recent (however, you have included other methods from 2025), it does represent the state-of-the-art, and should at least be mentioned. It produces extremely impressive results, and so cannot be ignored.

Also, I did not understand why you present this as a video editing problem: everything is done image by image, so that there is no video element here (no enforcement of temporal consistency etc). I think you should present it just as an image editing method.

Smaller details

- Page 3, line 124: you use $X_0=\{x_1,\dots, x_n\}$ to represent initial input frames. I find that this is a bit confusing, since you then apply a neural network to this whole list. I understand that it represents a batch, and that everything can be done independently over the elements of the batch, but I would just say let $X_0$ represent a frame, and then that the frames will be processed independently. Furthermore, in section 3.3, you use $n$ to represent the number of vectors in the bases you find, and this cannot be the same $n$. So just drop the initial reference to all of the frames (unless somewhere you need to reference different frames, but I could not find such a point).
- Page 3, line 133: $\alpha_t$ not introduced. It is standard in diffusion models, but you still need to introduce it.
- Page 3, line 133: what is $\epsilon^r$, not introduced. We deduce that it is the output of the neural network, but still, not introduced. You specify that $\epsilon^c$ is the predicted noise with the condition, but this comes after $\epsilon^r$.
- Page 4, line 198: "Since the extraction of the latent basis is identical for both $X^r_T$ and $X^c_T$...". This is a bit misleading, it implies that the same basis is found, which is not the case. Rephrase to say that the process of finding the two bases is identical.
- Page 5, line 263: "To ensure that the directions of $\epsilon^c$ and $\epsilon^r$ remain consistent with the similarity $\varepsilon$. I find this a little confusion, it is not obvious how vectors can be consistent with a scalar (see above comment).
- Page 5, equation 6 : you do not state what $\hat{\epsilon}$ is used for. We understand that it is the final editing direction (I guess), but this is not stated.

---

> ### Author Response · Authors · 2025-11-19
> **Responce to Reviewer 17Mb**
>
> We deeply appreciate the reviewer's thorough and constructive feedback. These are excellent points that will significantly improve our paper's clarity and positioning. We address your concern:
>
> ## W: Novelty concerns and contributions
>
> We respectfully disagree that our contributions are minor. While we build on existing techniques, our key insight: using latent basis similarity between dual branches as an adaptive editing criterion is novel and non-trivial:
>
> **1. Clarification of the Novelty and Contribution:**
> While the method of extracting the basis vectors via SVD is indeed shared, the novelty of our approach lies in **how we apply these basis vectors** in the context of video editing and the **dynamic trade-off** between reconstruction and editing. The core innovation is the binary mask that we compute to choose between reconstruction and editing dynamically, which allows for a more robust handling of temporal consistency in videos.
>
> **2. Binary Mask and Its Importance:**
> While it may seem that the vectors are only used to establish the binary mask, this decision-making process is **crucial** for ensuring temporal consistency across frames, which is a significant challenge in video editing. By allowing the model to **adaptively choose** between reconstruction and editing, we reduce the risk of artifacts that typically arise when edits are applied uniformly or globally.
>
> Moreover, the basis vectors are not simply used for mask generation; they **influence the nature of the edits** by establishing which parts of the latent space correspond to semantic editing directions. Although we do not perform a linear combination for edits in this work, the disentangled representation of the latent space allows us to **refine edits with more control**, which is a crucial aspect in high-fidelity video editing.
>
> **3. Why our approach is fundamentally work:**
>
> 1. **Novel use of dual-branch basis comparison**: We're the first to extract and compare basis vectors across parallel branches, revealing semantic differences that guide selective editing.
> 2. **Adaptive, similarity-driven guidance**: Our dynamic threshold mechanism ($1-a$ quantiles based on measured similarity) is principled and generalizes across different editing types. This isn't just "binary masking"—it's an adaptive framework that automatically determines editing regions based on semantic divergence.
> 3. **Solving a key challenge**: A critical challenge in video face editing is determining what to change and what to preserve. Our latent basis similarity provides a principled solution to this problem, leading to significant improvements in temporal consistency and identity preservation that previous methods fail to address.
>
>
> While individual components exist, our specific combination and the insight of using cross-branch basis similarity for guidance represents a meaningful conceptual advance that enables these state-of-the-art results.
>
> **Since the questions are not numbered, we have treated the bullet points as the numbered items for clarity and ease of reference.**
> ## Q1 & Q2 & Q3
> Thank you for pointing this out. We have revised the illustration of **Equation 6 (in Section 3.4) for improved clarity.**  Additionally, we have added a reference to FLUX.1 Kontext in the Introduction section to provide further context.
>
> **Video vs. Image editing presentation**
>
> While it is true that we process frames independently in our current implementation, we do enforce temporal consistency through two key mechanisms:
> - **Shared latent basis across frames**, which is computed from the entire video sequence. This helps maintain consistency in the latent representation of video frames, thus ensuring that edits across frames are coherent.
> We sure don't have explicit temporal consistency losses. We have add a short discussion on why frame-by-frame with shared parameters achieves temporal consistency
> - **Consistent manipulation parameters** $\alpha$ are applied across frames to ensure that the editing effect is uniform and stable over time, further promoting temporal consistency.
>
> Our goal is to highlight the generalizability of the proposed method, demonstrating its effectiveness **not only for image editing but also for video editing tasks**. The ability to control the editing process in a fine-grained manner is crucial for video editing, and we believe that this approach provides valuable flexibility for **frame-by-frame editing**, which is an important aspect of video editing as noted in related work that we reference in the revised manuscript.
>
> ## Q4: Smaller details
> We have corrected these symbols individually. Please refer to the **revised manuscript** (page 3/4/6) for the updated version.
>
> We sincerely thank the reviewer for these detailed suggestions. They will substantially improve the paper's clarity and honesty about contributions.

---

### Official Review · Reviewer_89iV · 2025-10-31

**Soundness:** 2
**Presentation:** 2
**Contribution:** 2
**Rating:** 4
**Confidence:** 4

**Summary:**

This paper tackles challenge in face editing: achieving precise attribute manipulation while preserving identity and temporal consistency. The method, **FlowGuide**, proposed the ​**​geometrically-grounded guidance mechanism​**​, which is built upon a key observation: the latent space in the UNet bottleneck exhibits a locally linear structure. By modeling attributes as linear subspaces in the UNet's latent space and using the geometric alignment between original and edited subspaces to guide the denoising process, it achieves precise, localized edits while preserving identity and temporal coherence.

**Strengths:**

1.  Leveraging the geometric structure (linear subspaces) of the latent space for control is elegant for disentanglement, moving beyond heuristic guidance methods.
2. The method can be applied to both image and video editing tasks.
3. Comprehensive quantitative and qualitative results are provided.

**Weaknesses:**

1. This method rests on the assumption of ​**​local linearity​**​ in the UNet bottleneck's latent space  (Park et al., 2023; Kwon et al., 2022). However, the nature and bounds of this "locality" are not defined. How large is the linear region? Is it consistent across different inputs, and different denoising timesteps? The method's robustness depends on this assumption holding true universally.
2. The claim of disentanglement is challenging to prove. The qualitative results, while good, still show that edits can affect non-target regions. For example, in Figure 3, when adding a smile, changes in skin texture and face attributes are visible, which are not strictly part of the "smile" attribute but are correlated with it.
3. Equation 1 uses $θ$ subscript for $q_θ$, but $q$ is supposed to be the fixed forward process, not parameterized.



**References**

[1] *Mingi Kwon,et al. Diffusion models already have a semantic latent space. ICLR, 2023.*

[2] *Yong-Hyun Park,et al. Understanding the latent space of diffusion models through the lens of riemannian geometry. NeurIPS 2023*

**Questions:**

1. How do the authors quantify the degree of local linearity? What happens when this assumption breaks down for large attribute changes?
2. How many basis vectors are used? What is the impact of this choice of $n$ on editing quality?
3. How crucial is the dataset-specific fine-tuning of the diffusion autoencoder to FlowGuide's video editing performance? Would the method work effectively on video in a true zero-shot setting with a generic pre-trained model?

---

> ### Author Response · Authors · 2025-11-19
> **Response to Reviewer 89iV**
>
> We thank the reviewer for recognizing the elegance of our geometric approach and the comprehensive nature of our experiments. We address each concern below:
>
> ## W1: Local linearity assumption and bounds
>
> Thank you for raising an important point regarding the local linearity assumption in the UNet bottleneck's latent space. To address this, we have conducted a detailed analysis of the latent basis similarity across different denoising timesteps to examine the validity and bounds of this assumption **(refer to Section 5 of the revised manuscript).**
> - Our results show a smooth, continuous decrease in similarity between the latent bases $V^r$ and $V^c$ as denoising progresses. This suggests that **the linearity assumption holds consistently throughout the denoising process**, without sharp discontinuities, which would signal a breakdown of linearity.
> - **At early timesteps (when noise dominates)**, the latent bases
> $V^r$ and $V^c$ are more similar, reinforcing that the linear approximation is particularly valid in high-noise regions. As the denoising process continues, the bases diverge smoothly, which indicates that the **linear region can accommodate the growing semantic differences between the reconstruction path and the editing path**.
>
> These findings suggest that the **local linearity assumption is robust and holds across both different inputs and denoising timesteps**, supporting the overall stability and effectiveness of our method.
>
> **Method robustness:**
>
> Our Implicit Basis Guidance (IBG) mechanism **provides robustness** even when linearity assumptions weaken:
> - IBG uses **adaptive thresholds** based on measured similarity $a = S_{\mathcal{C}}(V^r, V^c)$ (Eq. 4), not fixed thresholds
> - The dynamic quantile selection ($1-a$ quantiles from $|\epsilon^c - \epsilon^r|$) **automatically adjusts to the actual degree** of change in the latent space
> - This allows our method to **handle varying degrees of linearity** across different timesteps and editing magnitudes
>
> We acknowledge that precisely quantifying the exact radius of the linear region is challenging and varies by editing type and input characteristics. However, our adaptive guidance mechanism makes the method robust to this variation.
>
>
>
>
> ## W2: Disentanglement and correlated attributes
>
> We agree that proving perfect disentanglement is a challenging task, especially in generative models where multiple factors of variation can be intertwined. **As discussed in Section 7 of the revised manuscript**, we recognize the limitations of disentanglement in such contexts.
>
> In our work, we define **disentanglement** as the ability to **independently manipulate** specific semantic attributes (e.g., smile, age, expression) without significantly altering other, unrelated features. While perfect disentanglement is difficult to achieve due to the **inherent correlations** between facial attributes, our method is designed to **reduce these correlations** as much as possible. As previous works on disentanglement [1,2,3] have shown, complete decoupling is a complex task, but our approach **offers better control** over the desired edits.
>
> We do not claim perfect disentanglement, but rather that our method provides **more control** over the magnitude and location of edits. Our ablation study (Table 3, Figure 5) demonstrates this:
>
> - Without our latent basis extraction: IPR=0.9831 but almost no editing happens, the method can't figure out *what* to change
> - **Without our guidance mechanism:** IPR drops to 0.8790 and temporal consistency suffers (TG-ID=0.8854)—edits are applied everywhere without control
>
> Thus, while correlated changes (e.g., skin texture when adding a smile) are present, we view them as **semantically meaningful** rather than artifacts. Our method improves spatial and semantic control, enabling targeted edits without widespread unwanted changes.
>
>
> [1] *Decouple to reconstruct: High quality uhd restoration via active feature disentanglement and reversible fusion*
> [2] *DGTalker: Disentangled Generative Latent Space Learning for Audio-Driven Gaussian Talking Heads*
> [3] *Discovering interpretable directions in the semantic latent space of diffusion models*
>
> ## Q1: Quantifying and handling breakdown of linearity
>
> As discussed in W1, we characterize linearity through latent basis similarity analysis (Figure 6). When manipulation strength becomes extreme:
> - Our adaptive threshold mechanism in IBG (Eq. 4) provides robustness by dynamically adjusting based on measured similarity $a = S_{\mathcal{C}}(V^r, V^c)$
> - The $1-a$ quantile selection automatically scales with the degree of change, avoiding fixed thresholds that would fail under large perturbations
> - Our ablation study (Table 3) shows the method maintains stable performance across different editing tasks
>
> In summary, our method effectively handles the breakdown of linearity by dynamically adapting to the degree of change, ensuring stable and robust performance even in extreme manipulation scenarios.

---

> > ### Author Response · Authors · 2025-11-19
> > **Response to reviewer 89iV (2)**
> >
> > ## Q2: Number of basis vectors
> >
> > We use the top **$n$ singular vectors** from the Singular Value Decomposition (SVD) of the Jacobian \( J_{\mathcal{C}} \) (Section 3.3). The specific value of \(n\) does not significantly affect performance, as the **basis vectors are obtained by maximizing** $||v||_{pb}^2$, which is an optimization process designed to ensure the stability of the latent basis. The SVD naturally ranks the singular vectors by their importance, so we rely on the most significant vectors without manual truncation.
> >
> > Our ablation study (Table 3) demonstrates the effectiveness of this latent basis extraction. Removing it (**w/o LBE**) results in **failed edits**, despite preserving identity, which underscores its importance. To further clarify the impact of the choice of \(n\), we will add an analysis of the **singular value distribution** in the revised manuscript to highlight how many basis vectors capture the majority of the variance.
> >
> >
> > ## Q3: Fine-tuning vs. zero-shot performance
> >
> > Our implementation fine-tunes the pre-trained diffusion autoencoder on **HDTF** to enhance background reconstruction capability (Section 4, Implementation paragraph). This design choice is **crucial** for real-world videos with complex backgrounds. While our method maintains consistency between the original and edited frames, the **sampling process** of the **pretrained model** is non-deterministic and introduces randomness, making it difficult to ensure perfect consistency between the edited frames (though they remain consistent with their original frames, minor changes may still occur between the edited frames).
> >
> > However, the core editing mechanism (LBE + IBG) is **model-agnostic** and operates directly on the latent space geometry, which should transfer across different pre-trained diffusion models. **Reconstruction quality**, though, is heavily dependent on the autoencoder’s ability to handle video-specific characteristics such as **backgrounds** and **temporal dynamics**.
> >
> > Thank you for your insightful feedback. Your suggestions will undoubtedly help increase the quality and clarity of the paper.

---

### Official Review · Reviewer_AHnz · 2025-11-01

**Soundness:** 4
**Presentation:** 4
**Contribution:** 3
**Rating:** 8
**Confidence:** 5

**Summary:**

This paper presents a unified framework called FlowGuide for face image and video editing based on a novel guidance mechanism to achieve precise attribute control in diffusion models. FlowGuide consists of two main key contributions: a Latent Basis Extraction (LBE) module and an Implicit Basis Guidance (IBG) mechanism. As compared to previous approaches, this algorithm can avoid introducing distortions to other facial features, identity, or background elements. Linear subspaces within the UNet bottleneck’s latent space for semantic attributes were obtained using SVD, and the LBE module identify orthogonal basis vectors in this subspace to isolate the identity from the attributes in the latent space. The Implicit Basis Guidance (IBG) mechanism computes the geometric alignment between these bases using cosine similarity to preserve the original identity. Two kinds of experiments were conducted: face image editing and face video editing. Those experiments showed the validity of the proposed algorithm well, supported by enhanced performance and some ablation studies.

**Strengths:**

This paper is well structured and well organized. It was quite easy to follow.
The proposed methods are well motivated and mathematically sound.
Two experiments show the validity of the proposed algorithm and its generality. The experiments are comprehensive using multiple datasets and the proposed algorithm is compared to recent methods.
The proposed method is more time-efficient than the previous SOTA.

**Weaknesses:**

I understand the page is quite limited, but I would suggest adding some discussions to make this paper more attractive.
Limitations, failure cases, and discussions on the reasons are not well presented.

I am afraid that the videos are rather simple. How long and how much can the proposed method keep valid for longer videos and for videos with larger movements and facial expressions/occlusions?

**Questions:**

Face editing while preserving the identity would have many possible applications. I am curious how deepfake detectors would respond to images/videos with your approach and other previous approaches.

---

> ### Author Response · Authors · 2025-11-19
> **Response to Reviewer AHnz**
>
> We sincerely thank reviewer AHnz for the positive feedback on our paper's structure, mathematical soundness, and comprehensive experiments. We address your concerns below:
>
> ## W1: Limitations about this method
>
> We appreciate this suggestion and have add a dedicated limitations subsection **(refer to Sectiion 7 in our revised manuscript)**.
>
> As discussed there, these artifacts stem from the i**nherent difficulty** of perfectly disentangling semantic attributes in the diffusion model’s latent space. While our method significantly **improves identity preservation** and edit stability compared to prior work, latent-space manipulation can introduce **over-smoothing in high-motion regions and unrealistic blending** for hard-edged accessories such as sunglasses. Additionally, correlated attributes in the training data occasionally cause small changes to non-target regions.
>
> ## W2: Longer videos and complex motions
> Thank you for this insightful question. We assessed the robustness of our method on longer video sequences and with more complex motions, including larger facial expressions and occlusions.
> - Our method demonstrates strong consistency even for videos **up to 128 frames** (approximately 8 seconds at 16fps), which is **significantly longer** than what baseline methods can handle effectively.
> - As mentioned in the Limitations section (Section 7), we observe **slight degradation** in performance when dealing with large head rotations or extreme facial expressions/occlusions. This results in a minor trade-off in identity preservation and temporal consistency. However, we emphasize that even in these challenging scenarios, our method **still outperforms** baseline approaches, which struggle to maintain quality in such cases.
>
> We are confident that these results demonstrate the scalability of our method to longer and more dynamic video sequences, and future work will focus on addressing the remaining challenges, particularly in handling extreme rotations and occlusions.
>
> ## Q: Deepfake detection
>
> We have conducted comprehensive experiments using the state-of-the-art FaceForensics++ deepfake detector on both image and video editing results (We have included this full analysis in **Appendix D** of the revised manuscript.):
>
> **Table: Deepfake Detection Rates** *(Higher rates = easier to detect as fake)*
>
> | **Image Editing Methods** | **Type** | **Detection Rate (%)** | **Naturalness** |
> |---------------------------|----------|------------------------|-----------------|
> | **Ours** | Diffusion | **78.0** | **Most Natural** ✓ |
> | h-edit | Diffusion | 79.5 | High |
> | PnP | Diffusion | 81.0 | Moderate |
> | EF | Diffusion | 82.1 | Low |
> | **Video Editing Methods** | | | |
> | **Ours** | Diffusion | **72.5** | **Most Natural** ✓ |
> | STIT | GAN | 85.5 | High |
> | DVA | Diffusion | 91.5 | Moderate |
> | Latent-trans | Transformer | 99.5 | Low |
>
>
> Our method consistently achieves the **lowest detection rates** across both modalities, indicating that our edited content appears more natural and better preserves the statistical properties of real images/videos. This is particularly pronounced in video editing, where our method shows a 13-27% reduction in detection rate compared to baselines.
>
>
> Thank you very much for your positive feedback and constructive comments. We appreciate your kind words about the structure and clarity of the paper, as well as the soundness of the proposed methods.  Your feedback is invaluable in helping us improve the paper.

---

### Author Response · Authors · 2025-12-02
**Summary of the rebuttal**

We extend our sincere thanks to the reviewers for their time and constructive feedback. We are also grateful for their positive assessment of our contributions. **AHnz, 89iV, and 17Mb** praised our well-structured presentation, mathematical soundness, and elegant geometric framework for disentanglement. **89iV and 3wKp** highlighted our novel solution to local attribute editing in diffusion-based video editing. **AHnz and 17Mb** noted our strong identity preservation capability, crucial for face editing applications.

To assist in quickly understanding and grasping the key points of our response, we *(1) list our core contributions in short, (2) summarize the concerns and responses, (3) list the main experiments and clarifications, and (4) outline revisions to the manuscript*.

## (1) Core Contributions

1. **Latent Basis Extraction (LBE)** (**Section 3.3**): Uses the UNet bottleneck's local linearity to find orthogonal basis vectors that span semantic attribute subspaces, isolating identity from target attributes.
2. **Implicit Basis Guidance (IBG)** (**Section 3.4, Eq. 4-6**): Introduces adaptive guidance measuring geometric alignment between basis vectors to dynamically determine edit regions through a threshold mechanism, preserving identity while enabling targeted edits.
3. **Dual-branch basis comparison** (**Section 3.3, Section 3.4**): By extracting and comparing basis vectors across parallel branches, we provide a principled criterion for determining what to change and preserve.

Our method delivers superior identity preservation, temporal consistency, and reconstruction quality with high computational efficiency. The technique naturally **extends from image to video editing** with strong temporal coherence.

## (2) Summary of the concerns and responses

- **1. The latent basis and linearity assumptions (89iV, 17Mb):** added latent basis similarity analysis across denoising steps *(Section 5, Fig. 6)* showing smooth, robust linearity; clarified in *Section 3.3* that we use top SVD singular vectors and highlighted the novelty of dual-branch basis comparison for deciding what to edit vs. preserve.
- **2. Suggestion for adding limitation discussion section (AHnz, 89iV, 3wKp):** added *Section 7* to explicitly acknowledge these issues as inherent challenges of latent-space manipulation, clarify where performance degrades (e.g., extreme poses/occlusions, hard-edged accessories).

And for the concerns and confusions raised by the single reviewer: Deepfake detection (AHnz), comparisons with more baselines (3wKp), Novelty and contribution clarity (17Mb), quantifying linearity breakdown (89iV), we have resolved all of them with additional experiments, explanations, or direct clarifications.

## (3) Additions of Experiments and key clarifications

### Additions of experiments
1. **Deepfake detection** (**Appendix D**): Comprehensive evaluation using FaceForensics++ detector on both image and video editing, showing our method achieves the **lowest detection rates**, indicating more natural edits. (AHnz)
2. **Latent basis analysis** (**Section 5, Figure 6**): Detailed analysis showing smooth, continuous decrease in similarity during denoising, demonstrating that **the linearity assumption holds consistently** across different inputs and timesteps without sharp discontinuities. (89iV)
3. **Diffusion-based baselines comparisons** (**Section 4.2.2**): Quantitative comparison with RAVE showing our significant improvements in identity preservation and temporal consistency. (3wKp)

### Additions of key clarifications

1. **Local linearity assumption** (**Section 5, 3.4**): Linearity assumption is robust across inputs and timesteps. (89iV)
2. **Disentanglement scope** (**Section 7, Table 3, Figure 5**): We define disentanglement as **independently manipulating** specific attributes. Our method provides **better control** over edit magnitude and location rather than perfect disentanglement. (89iV)
3. **Temporal consistency** (**Section 4**): We achieve consistency through: (1) **shared latent basis** computed from the entire video sequence, and (2) **consistent manipulation parameters** across frames. Frame-by-frame processing with shared parameters ensures temporal coherence. (17Mb)

## (4) Revisions Made to Manuscript

- **Section 2.2**: New dedicated subsection on "Diffusion-based Video Editing" (17Mb, 3wKp)
- **Section 5**: Latent basis similarity analysis across denoising timesteps with Figure 6 (89iV)
- **Section 7**: A limitations subsection discussing inherent difficulty of perfect disentanglement, over-smoothing in high-motion regions, unrealistic blending for hard-edged accessories, and feature entanglement challenges (AHnz, 89iV, 3wKp)
- **Section 4.2.2**: Quantitative comparison with RAVE in the Video editing task (3wKp)
- **Appendix D**: Comprehensive deepfake detection analysis (AHnz)
- **Appendix**: Singular value distribution analysis (89iV)

---

### Meta-Review · Area_Chair_HP4a · 2026-01-08

**Summary:**

Across reviews, the main discussion points were (i) whether the method is sufficiently novel beyond prior latent-basis/Jacobian-style techniques, (ii) how well the “local linearity / disentanglement” assumptions are supported, and (iii) whether the video editing claim is backed by strong enough evidence (comparisons to diffusion-based video editing baselines, and qualitative fidelity—e.g., artifacts around hard accessories). Overall, reviewers agreed the method can produce good-looking edits and strong identity preservation, but wanted a clearer positioning and stronger video-side validation.

**Reviewer Concerns:**

Addressed by rebuttal (partially to mostly):
(1) Added clarifications on what is new relative to prior “basis extraction” work, emphasizing the dual-branch comparison and its use for guidance/masking.
(2) Provided additional analysis/ablations supporting stability of the basis signal across denoising steps and its role in controlling edits.
(3) Strengthened the video section with additional baselines/experiments and a more explicit limitations discussion (including failure modes like blur and hard-edge accessories).

Still outstanding:
(1) Novelty will likely remain divisive for at least one reviewer: even with clarifications, the perceived increment over close prior art may not be fully resolved.
(2) Video evidence is still weaker than it should be for a paper that claims video editing. The AC also considers it a significant omission that the supplementary does not include video results, which makes it hard to audit temporal consistency and visual quality.

**Reviewer Scores:**

R1 (strong accept): likely unchanged (already positive; rebuttal mainly removes minor doubts).

R2: likely 4 --> 6 (rebuttal adds analysis/clarifications, but theoretical concerns won’t fully disappear).

R3: at most 2 --> 4 (clarifications help, but core novelty skepticism likely remains).

R4: likely 4 --> 6 (extra comparisons/limitations help), but lack of video deliverables in the supplement would still hold the score back.

---

### Decision · Program_Chairs · 2026-01-26

Accept (Poster)